# Biomass removal promotes plant diversity after short-term de-intensification of managed grasslands

**Karl Andraczek**[1]*, **Alexandra Weigelt**[1,2], **Judith Hinderling**[3‡], **Lena Kretz**[1‡], **Daniel Prati**[3‡], **Fons van der Plas**[1,4]

**1** Department of Life Sciences, Systematic Botany and Functional Biodiversity, University Leipzig, Leipzig, Germany, **2** German Centre for Integrative Biodiversity Research (iDiv) Halle-Jena-Leipzig, Leipzig, Germany, **3** Institute of Plant Sciences, University of Bern, Bern, Switzerland, **4** Plant Ecology and Nature Conservation Group, Wageningen University, Wageningen, the Netherlands

☯ These authors contributed equally to this work.
‡ JH, LK and DP also contributed equally to this work.
* karl.andraczek@uni-leipzig.de

**Data Availability Statement:** Data and Code are available in the BExIS database of the Biodiversity Exploratories program https://www.bexis.uni-jena.de/ddm/publicsearch/index (ID 31370, DOI: https://

## Abstract

Land-use intensification is one of the main drivers threatening biodiversity in managed grasslands. Despite multiple studies investigating the effect of different land-use components in driving changes in plant biodiversity, their effects are usually studied in isolation. Here, we establish a full factorial design crossing fertilization with a combined treatment of biomass removal, on 16 managed grasslands spanning a gradient in land-use intensity, across three regions in Germany. Specifically, we investigate the interactive effects of different land-use components on plant composition and diversity using structural equation modelling. We hypothesize that fertilization and biomass removal alter plant biodiversity, directly and indirectly, mediated through changes in light availability. We found that, direct and indirect effects of biomass removal on plant biodiversity were larger than effects of fertilization, yet significantly differed between season. Furthermore, we found that indirect effects of biomass removal on plant biodiversity were mediated through changes in light availability, but also by changes in soil moisture. Our analysis thus supports previous findings, that soil moisture may operate as an alternative indirect mechanism by which biomass removal may affect plant biodiversity. Most importantly, our findings highlight that in the short-term biomass removal can partly compensate the negative effects of fertilization on plant biodiversity in managed grasslands. By studying the interactive nature of different land-use drivers we advance our understanding of the complex mechanisms controlling plant biodiversity in managed grasslands, which ultimately may help to maintain higher levels of biodiversity in grassland ecosystems.

[doi.org/10.25829/](https://doi.org/10.25829/) BEXIS.31370-13). The raw dataset with the ID 24766 is available in the BExIS database under the URL [https://www.bexis.uni-jena.de/ddm/publicsearch/index](https://www.bexis.uni-jena.de/ddm/publicsearch/index). Raw data sets with the ID 31203, 31204 will be publicly available under the URL [https://www.bexis.unijena.de/PublicData/About.aspx](https://www.bexis.unijena.de/PublicData/About.aspx) from May 2023 on. The Data set with the ID 31180 will be publicly available under the URL [https://www.bexis.unijena.de/PublicData/About.aspx](https://www.bexis.unijena.de/PublicData/About.aspx) from December 2022 on. Until then, data is available upon request ([bexis@uni-jena.de](mailto:bexis@uni-jena.de)).

**Funding:** The work has been funded by the German Research Foundation (DFG) Priority Program 1374 'Biodiversity-Exploratories' (received by: F.v.P., BEF-Loops Nr.433266560, Exploratories project phase Nr. PL 891/3-1). The funders had no role in study design, data collection and analysis, decision to publish, or preparation of the manuscript.

**Competing interests:** The authors have declared that no competing interests exist.

# 1 Introduction

Grassland ecosystems cover almost a third of the global terrestrial surface and harbour a substantial amount of biodiversity [1]. At the same time, grassland biodiversity is threatened by various anthropogenic influences, including land-use intensification [2–4], which in central European grasslands mainly encompasses the intensification of mowing, grazing and fertilization frequency [5]. However, as land-use comprises a number of different drivers, including fertilization and mowing (or grazing), we need to understand the interactive effects of these drivers to counteract their effects on plant biodiversity loss [6]. Different land-use drivers typically covary, for instance, agricultural grasslands that are frequently mown are generally also heavily fertilized [6], but we usually study their effects in isolation. So far, only a few studies investigated their interactive effects in real-world settings [6, 7], especially in a scenario of land-use de-intensification (i.e. a reduction of land-use intensity, e.g. mowing frequency, grazing or fertilization intensity). Thus, we still a have limited understanding of how different land-use drivers can be optimized in order to maintain higher levels of plant biodiversity.

In grasslands, fertilization causes declines in plant biodiversity either directly or indirectly [6, 8, 9]. Following the Resource competition theory by Tilman et al. [10], the addition of a limiting resource alters interspecific competition, leading to only a few species to coexist. For example, Harpole et al. [11] found that the addition of multiple limiting resources caused plant biodiversity to decline, likely driven by reduced nutrient niche-dimensionality below ground. However, fertilization may also indirectly affect plant biodiversity via increasing biomass production of living and dead biomass, and hence also standing biomass, thereby altering light competition [8, 12, 13, but see 14], soil acidity, nutrient mineralization rates, or the activity of pathogens [15, 16]. Especially light competition is expected to be highest in summer, when standing biomass is highest, limiting reproduction and survival [17, 18]. Following that, the relative importance of indirect effects of fertilization on biodiversity mediated through changes in light availability likely differs between season. Furthermore, plant biodiversity loss driven by light competition is non-random, as slow-growing and small species are successfully outcompeted by fewer fast-growing and tall species which become dominant [12, 19, 20]. Hence, we would expect changes in both species richness but also Shannon diversity [17]. So far, there is evidence that both direct and indirect effects of fertilization are important in driving changes in biodiversity in grasslands [21].

While fertilization generally causes plant biodiversity to decline, mowing and grazing can induce more mixed responses depending on their intensity [6, 22]. Both, mowing and grazing, can promote plant biodiversity, although partly due to different mechanisms. Mowing at intermediate frequency can promote plant species richness and diversity by decreasing light competition, while concomitantly increasing competitive abilities (e.g. germination rates) of subdominant species [8, 23–26]. Furthermore, hay removal due to mowing decreases soil nitrogen pools [27, 28], thereby reducing nutrient availability in the soil and positively affecting plant biodiversity. Similar to mowing, grazing positively affects plant biodiversity under certain conditions [23, 29–31], by decreasing light competition as a consequence of biomass removal, as well as by the creation of microsites promoting seedling establishment [32, 33]. However, both mowing and grazing can also negatively affect plant biodiversity at higher intensities [6, 34, 35]. Although biomass removal may promote species richness by increasing light availability, it is possible that under dry conditions, seedling survival is reduced specifically in gaps due to higher drought stress [36, 37]. Hence, possible positive effects of biomass removal via increasing light availability may depend on soil moisture [14, 38]. As soil moisture is most limited in summer, the relative importance of indirect effects of biomass removal on biodiversity could also be expected to differ between growing seasons. Although there is a

large body of research investigating the effects of land-use on plant biodiversity in managed grasslands, there is still a limited understanding about the relative and potentially opposing influences of the different land-use drivers, i.e. fertilization and mowing/grazing.

Previous studies investigating effects of land-use on plant biodiversity [6, 13, 39] found confounding effects of different land-use drivers, which could not be disentangled, due to the lack of an appropriate study design. For instance, in managed grasslands mowing is often found to be correlated with fertilization intensity [40]. Furthermore, mowing and grazing may (partly) compensate for the generally negative effects of fertilization on plant biodiversity [12, 32, but see 41], by reducing litter accumulation and thereby reducing light competition [26], but also, and in the case of mowing, by removing substantial amounts of nutrients from of the system [42, 43]. While crossed fertilization and mowing/grazing studies exist, there are typically carried out within grasslands with initially low land-use intensity, but not in systems with a history of intensive grassland management. Furthermore, few studies manipulated the intensity of different land-use drivers within the same grassland, which provides crucial insights on how biodiversity responds to reductions of different land-use drivers in a realistic setting. Thus, to truly unravel the drivers of biodiversity loss in managed grasslands and to gain insights in potential biodiversity recovery, it is necessary to study the interaction of land-use drivers by using a suitable experimental design. Ultimately, comparing the strength of direct and indirect effects of different land-use drivers across different seasons may advance our understanding on the complex mechanisms which alter plant biodiversity in managed grasslands. In doing so, we may also improve our current knowledge on which land-use drivers should, or should not, be de-intensified if we want to promote plant biodiversity.

In this study, we test an hypothesis involving proposed effects of fertilization and biomass removal (by mowing/grazing) on plant biodiversity (here species richness and Shannon diversity). Specifically, we aim to distinguish between direct and indirect effects of fertilization on plant biodiversity, and how biomass removal due to mowing/grazing mediates these pathways (Fig 1). We thus aim to understand how different drivers of land-use de-intensification may, or may not, promote plant biodiversity in the short-term, while also addressing seasonal variability of these relationships. To address these questions, we designed a full factorial experiment, replicated within managed grasslands in three regions in Germany. We compare the effects of land-use drivers crossing fertilization with a combined treatment of biomass removal (mowing/grazing treatment): (a) fertilized & biomass removal, (b) unfertilized & biomass removal, (c) unfertilized & reduced biomass removal and (d) fertilized & reduced biomass removal. In particular, we tested the following hypotheses:

1. Fertilization will cause plant species richness and Shannon diversity to decline, both directly and indirectly, by altering light competition due to increased biomass production and thereby increased standing biomass.

2. Biomass removal partly compensates for the negative effects of fertilization on species richness and Shannon diversity, by decreasing levels of standing biomass and thereby reducing light competition or changing the competitive dominance of plant species.

## 2 Material and methods

### 2.1 Study area

We studied the interactive effect of fertilization and mowing/grazing on plant biodiversity in 16 commercially managed (unsown) grasslands located in three regions in Germany as part of the Biodiversity Exploratories project (www.biodiversity-exploratories.de; [44]). The regions

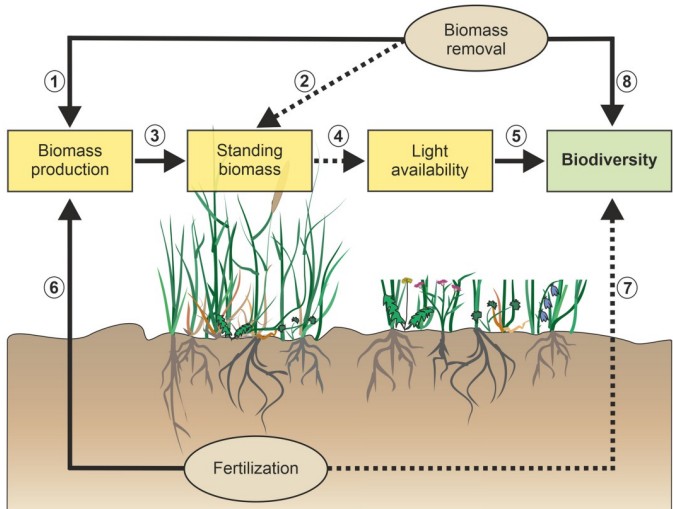

**Fig 1. Conceptual framework.** Path diagram showing how the different land-use drivers fertilization and biomass removal (mowing/grazing), may directly or indirectly (via increasing or decreasing light competition due to changes in biomass production/standing biomass) affect plant biodiversity (represented here by species richness and Shannon diversity) in managed grasslands. Pathways are sequentially numbered referring to supporting evidence, 1: [46, 47], 2: [10, 44], 3 and 4: [16], 5: [11, 15], 6: [47], 7: [13, 14, 48], 8: [28, 33]. Alternatively changes in soil moisture can mediate pathways between biomass removal or fertilization and plant biodiversity [17] (not shown here). For further information also on seasonal variability of relationships, see the introduction. Dashed arrows indicate negative and straight arrows positive relationships between two variables.

are distributed across the south-east (UNESCO Biosphere Reserve Schwäbische Alb—Swabian Jura), the centre (National Park Hainich-Dün) and the north-east of Germany (UNESCO Biosphere Reserve Schorfheide-Chorin). Due to various climatic and edaphic differences, these regions vary in several environmental conditions, such as temperature and soil fertility (deep layers of organic soils) being highest in north-west, or annual mean precipitation and elevation being highest in the south-west (see [44] for more details on regional differences). All 16 grasslands were commercially managed, spanning a gradient of background land-use intensity (LUI) which is a composite measure of mowing frequency, livestock units and amount of fertilization (see S17 Table). By including multiple regions and spanning a gradient in background grassland land-use intensity (rather than conducting a more conventional experiment in highly controlled settings), we aimed to investigate how the de-intensification of different grassland land-use drivers (fertilization vs mowing/grazing) interactively drive plant biodiversity in semi-natural systems. Across all studied grasslands, the "most 10 abundant" species were: *Poa pratensis* L., *Lolium perenne* L., *Dactylis glomerata* L., *Taraxacum sp.* F.H. Wigg., *Alopecurus pratensis* L., *Phleum pratense* L., *Trisetum flavescens* L., *Festuca pratensis* Huds., *Achillea millefolium* L., *Bromus hordeaceus* L.

## 2.2 Experimental design

In all three regions, we investigated how different land-use drivers (fertilization and mowing/grazing) influence plant biodiversity via direct or indirect pathways. In this study we focus on the grassland management components fertilization and mowing/grazing, as in commercially managed grasslands within central Europe, those are the most dominant land-use components representing important predictors for both plant biodiversity and biomass production [5, 6, 45]. To do so, we selected 16 managed grassland sites (6 in the Schwäbische Alb and Hainich-

Dün, and 4 in Schorfheide-Chorin) and set-up a full factorial experimental design, by reducing the intensity of biomass removal and fertilization in marked treatments from autumn 2019 onwards. Thus, we established four 7×7 m plots per grassland plot, containing one of four combinations: fertilized & biomass removal (+F+R), unfertilized & biomass removal (-F+R), unfertilized & reduced biomass removal (-F-R), fertilized & reduced biomass removal (+F-R) (S1 Fig). In the +F+R treatment the amount of fertilization (0.57–7.43 kg N m$^{-3}$ year$^{-1}$), grazing (number of livestock units [LU] per grazing duration, 0.02–5.69 LU * day ha$^{-1}$ year$^{-1}$) and mowing (1.02–2.73 number of cuts per year) differed along a land-use gradient (S16 Table). In the -F+R treatment, fertilization was stopped completely, while mowing and grazing continued at the same intensity and frequency as in +F+R plots. In the -F-R treatment, land-use intensity was reduced to mowing only once a year in August or September, while stopping grazing and fertilization completely. This disturbance (here by reduced mowing regime) was done, because some level of minimal disturbance is required in central European grasslands, to prevent shrub encroachment and forest succession [46]. As in the reduced land-use treatment, land-use intensity was reduced in the +F-R treatment, while fertilization (organic or inorganic fertilizer depending on fertilizer type used by farmer) was manually applied to avoid cross-contamination with fertilizer between adjacent treatments. The amount and frequency of fertilizer application was comparable to the amounts of fertilizer applied in the respective fertilized & biomass removal plot. For each treatment (i.e. +F+R, +F-R, -F+R, -F-R), there was one plot in 16 different grasslands (with one subplot located in each plot), yielding to 64 studied plots in total.

## 2.3 Data collection

All data was collected in the spring and summer season, at peak biomass (April to May and July to August respectively), in 2020 and 2021. In each season, we estimated the vascular plant species specific cover in each subplot. Additionally, in each subplot we assessed standing biomass by using a rising plate meter (Jenquip Manual Plate Meter), as the mean of four measurements per subplot. By measuring vegetation height and density, derived from the height above ground level at which the disc (Ø 35.5 cm, 384 g) is supported by the vegetation, the rising plate meter can be used to estimate standing biomass non-destructively [47]. Biomass estimates derived from the rising plate meter were calibrated using data from two additional 1×1 m subplots (100 calibration subplots per region, 300 in total) in each grassland plots (containing +F+R subplots) and in plots within comparable managed grasslands differing in land-use intensity (see BExIS dataset ID 31180). In these calibration plots, during spring 2021, both biomass estimations using the rising plate meter and actual biomass measurements (clipping and drying above ground vegetation) were performed. To quantify light competition for plants below the canopy, we measured light availability of photosynthetically active radiation (using PAR Sensor, Skye Instruments, in µmol sec$^{-1}$ m$^{-2}$) both at ground level and at 1.5 m height. To test an alternative indirect relationship of fertilization or biomass removal via soil moisture, we additionally measured soil moisture (ML3 ThetaProbe Soil Moisture Sensor, in %) with three replicates per subplot. In order to account for biomass that has been removed due to mowing or grazing when calculating biomass production for the summer season, we further used data on grazing duration, livestock units per area and hay yield of mowing derived from yearly standardized questionnaires of land-owners and farmers (for more information see 48), which we obtained from the database BExIS (see BExIS dataset ID 26487, http://doi.org/10.17616/R32P9Q). In spring, biomass production was considered equal to standing biomass in spring, as most grasslands were hardly grazed or not mown before our biomass measurements took place (except for one grassland). Additionally, we quantified background fertilization intensity

(kg N ha$^{-1}$ year$^{-1}$), background grazing intensity (Livestock units $*$ day ha$^{-1}$ year$^{-1}$) and background mowing intensity (cuts year$^{-1}$), for each of the years 2017–2019 (before our experiment was set up) also derived from yearly std. questionnaires [48]. Background intensity of all land-use components were calculated via the LUI calculation tool [49] implemented in BExIS (http://doi.org/10.17616/R32P9Q).

## 2.4 Data processing and calculations

All species names were taxonomically standardized according to the accepted species names in The Plant List (www.theplantlist.org, using the TAXONSTAND R package [50]. For each sub-plot, we quantified species richness and the effective Shannon Diversity [51, 52]. Effective Shannon Diversity was calculated as:

$$H' = \exp\left(-\sum_i p_i \log(p_i)\right) \tag{1}$$

Standing biomass was quantified using biomass estimates which were converted into standing biomass using a calibration formula. This calibration formula was based on calibration data collected in spring 2021 from 2 x 150 subplots (1×1 m each) in additional grassland sites in all three regions (2 x 50 subplots per regions) including all +F+R plots, containing biomass estimates measured via a rising plate-meter (average per subplot) and actual standing biomass measurements from the same subplot. The calibration formula was then derived by performing a linear model with actual standing biomass (in g m$^{-2}$) as response variable and rising-plate meter measurements (in 0.5 cm increments) as predictor variable. Prior to calculating the calibration model, we excluded data from five subplots due to measurement errors. Following the calibration model, biomass estimates explained 83.22% of the variance in actual standing biomass (t = 37.94, p > 0.01, RSD = 38.85 g m$^{-2}$, S18 Table). Using the intercept and slope of the calibration model, we converted the biomass estimates into actual standing biomass following the equation: dry biomass (g m$^{-2}$) = -38.66 + 8.68 × Plate meter measurement. In addition to standing biomass, we also quantified biomass production in all subplots which was defined as all biomass produced in a given time (Spring until Summer) including biomass removed by mowing or grazing. Only biomass removed by higher trophic levels other than livestock (e.g. herbivorous arthropods) could not be included in our biomass production estimate. We quantified biomass production following Riehl [53]:

Biomass production $_{Summer}$ =

(Standing biomass$_{Summer}*$(Livestock units $_{Summer}*$Grazing duration$_{Summer}*$14700) + Yield mowing$_{Summer}$)$-$ $\tag{2}$

(Standing biomass$_{Spring}*$(Livestock units$_{Spring}*$Grazing duration$_{Spring}*$ 14700) + Yield mowing$_{Spring}$)

with livestock units in unit m$^{-2}$, grazing duration in number of grazing days, '14700' referring to the mean fodder consumption of an average livestock unit in g per day [53], and yield mowing in g m$^{-2}$. In particular, yield mowing is defined for spring, as the sum between all mowing events between the start of the growing season (1$^{st}$ of April) and the end of the first grassland survey (15$^{th}$ of May), and in summer, as the sum for all mowing events between the end of the first grassland survey and the end of the second grassland survey (14$^{th}$ of August). These periods were also applied to calculate the grazing duration and livestock units per subplot. We only calculated biomass production in summer as in this season we expect a mismatch between standing biomass and biomass production due to land-use (e.g. removal of biomass due to mowing or grazing), while in spring biomass measurements were in most plots done before mowing and grazing events took place, and hence standing biomass was almost

equivalent to biomass production of the early spring season. To quantify light competition in each subplot we calculated the inverse ratio between the average light availability of photosynthetically active radiation above canopy height and ground level (light availability hereafter, light availability increases from 0 to 1) per subplot. Additionally, to obtain a mean background fertilization intensity value over the time before our experiment was set up (2017–2019), background fertilization intensity was averaged across time.

## 2.5 Data analyses

All data analyses were performed using the statistical software R v. 4.1.1 [54]. Statistical analyses were performed using the packages lme4 [55], lmerTest [56], stats4 [54], multcomp [57], car [58], MuMIn [59] and PIECEWISESEM [60].

**2.5.1 Overall effect of treatments on plant species richness and diversity.** To study how the different treatments (i.e. +F+R, +F-R, -F+R, -F-R) affected species richness, Shannon diversity, standing biomass or biomass production, we performed linear mixed models including treatment and region as fixed effects and grassland as a random effect. To test for pairwise comparisons between the different treatments we estimated least-square means and computed contrasts using the function emmeans from the package emmeans [61]. For all models we performed a forward model selection procedure. Specifically, we started with a model without fixed factors and we then stepwise added treatment and region as additional predictors if these i) did not increase the AIC and ii) did not exceed a variance inflation factor (VIF) of 3 [62], using the vif function from the package car. That model selection procedure allowed us to select the most parsimonious model while avoiding multicollinearity between predictors. Importantly, even when not being part of the most parsimonious model, we always included treatment in our final model. Furthermore, if both treatment and region were part of the most parsimonious model, we tested for an interaction between these, by comparing the AIC of the final model and the model with an interaction term.

**2.5.2 Treatment-mediated changes in community composition.** To test how the different treatments induced changes in plant community composition, separately for each region and season in 2021 (for results on 2020 see S10 Fig), we used a non-metric multidimensional scaling (NMDS) analysis, based on Bray–Curtis similarity index as an ordination technique using the function metaMDS from the package vegan [63]. Furthermore, to statistically test the dissimilarities between the species compositions of the different treatments after two years since the experiment was set up (in 2021), we performed a permutational multivariate analysis of variance (PERMANOVA), using the package pairwiseAdonis [64]. To visualize whether certain environmental factors explain shifts in plant community composition we further fitted environmental vectors (light availability, soil moisture and standing biomass) onto the NMDS ordination for all regions combined (for results on 2020 see S11 Fig). Correlation of environmental variables with the NMDS axes was tested using the envfit() function from the package vegan [63].

**2.5.3 Direct and indirect effects of land-use on species richness and diversity.** We expected that fertilization can both directly, but also indirectly (mediated via changes in biomass production, standing biomass and light availability) affect species richness and Shannon diversity. Furthermore, we expected biomass removal to mediate the indirect effect of fertilization on species richness/Shannon diversity by removing biomass and thereby decrease light competition. We additionally tested for an indirect relationship between fertilization and biomass removal via soil moisture. To guarantee comparability between spring and summer data, we used soil moisture as an additional indirect pathway in both spring and summer based models. To test these assumptions, we constructed a hypothesis driven causal model using

linear mixed-effect models within a PiecewiseSEM [60]. In our models, we converted the treatment ID into the dummy variables "fertilization", and "biomass removal", with "0" indicating absence and "1" presence of the respective land-use component. We further log-transformed light availability, to meet model assumptions regarding linearity. To statistically correct for the influence of year and sampling date we further constructed a linear model with each continuous variable (species richness, species diversity, biomass production, standing biomass and log light availability) as response, with year and sampling date as fixed effect and with an interaction term. Finally, we extracted the model residuals (without the explained variance by year and sampling date) which were used as input for all further SEM models.

We constructed four separate, *a priori* hypothesized models, testing the causal relationship between the land-use drivers fertilization and biomass removal with species richness or Shannon diversity in spring and summer separately. First, we ran the initial SEM models as a list of causal relationships. Secondly, we inspected all initial model results, goodness-of-fit tests and Fisher's C statistics, and, only if necessary, added predictors that significantly improved the AIC and the model fit with p-values higher than 0.05. During this process, for both summer models we compared competing models ex- or including an error correlation structure between biomass production and soil moisture (species richness model: AIC without correlated error = 93.3, with correlated error = 92.1; Shannon diversity model: AIC without correlated error = 95.1, with correlated error = 94.0). For spring models, no error correlation structure between biomass production and soil moisture could be tested, as these models did not include a biomass production variable. This was done in order to increase the goodness-of-fit of the models, while we deliberately did not include any causal relationship between biomass production and soil moisture since it was difficult to distinguish between cause and effect. To statistically correct for the confounding effects of covarying factors we included background fertilization intensity of the time before the experiment was set up (mean of 2017–2019) as covariate for models predicting species richness/diversity, standing biomass and biomass production. In all models, grassland was treated as a random factor nested within study region by using the lme function from the package nlme [65]. As the residual variance of light availability increased with increasing standing biomass, we included a power variance structure for the variance covariate standing biomass in all models predicting light availability, to account for heterogeneity in the residuals. We inspected the assumptions of normality of model residuals visually. PiecewiseSEMs were performed using the R package PIECEWISE-SEM [60].

## 3 Results

### 3.1 Overall effects of fertilization and biomass removal on plant species richness and species composition in different seasons

Overall, we did not detect a consistent effect of treatments on species richness (Table 1, S2 Fig). Similar, subplots with +F+R did not significantly differ from any other treatment (see results of pairwise differences in S2 Table). However, we generally observed a lower species richness in +F-R subplots compared to all other treatments (although not statistically significant). In the Hainich-Dün in spring 2021, species richness was found to be marginally significantly, lower in subplots which were fertilized but had reduced biomass removal compared to subplots with no fertilization but biomass removal (S2 Table). Our NMDS results showed no significant differences in community composition between the different treatments in neither of the regions, nor in any of the different seasons of the year 2021 (Fig 2, S7 Table).

**Table 1. Effects of unfertilized & biomass removal (-F+R), fertilized & reduced biomass removal (+F-R) and unfertilized & reduced biomass removal (-F-R) treatment on species richness.** Treatment effects indicate standardized effect sizes, with associated confidence intervals in squared brackets. P-values (P) indicate overall significant effect of treatments on species richness. If most parsimonious model included an interaction between treatment and region, we also show regional specific effect sizes. For more detailed information on statistical results see S1 Table. Note that rows are relative to the intercept (Alb and +F+R treatment).

| | | Std. effect sizes of treatments on species richness | | | | *P* |
|---|---|---|---|---|---|---|
| | | *all* | *Alb* | *Hai* | *Sch* | |
| **-F+R** | *SP '20* | -0.04 [-0.22, 0.14] | - | - | - | 0.46 |
| | *SU '20* | - | 0.14 [-0.18, 0.47] | 0.00 [-0.45, 0.17] | -0.18 [-0.49, 0.09] | 0.11 |
| | *SP '21* | - | 0.10 [-0.24, 0.44] | -0.19 [-0.62, 0.04] | -0.04 [-0.44, 0.17] | 0.74 |
| | *SU '21* | - | 0.25 [-0.05, 0.56] | 0.20 [-0.34, 0.24] | 0.03 [-0.49, 0.05] | 0.12 |
| **+F-R** | *SP '20* | -0.14 [-0.33, 0.04] | - | - | - | 0.46 |
| | *SU '20* | - | -0.14 [-0.47, 0.18] | -0.15 [-0.32, 0.30] | -0.27 [-0.41, 0.16] | 0.11 |
| | *SP '21* | - | 0.08 [-0.26, 0.42] | 0.00 [-0.41, 0.25] | -0.08 [-0.46, 0.14] | 0.74 |
| | *SU '21* | - | -0.04 [-0.34, 0.27] | -0.14 [-0.39, 0.19] | -0.13 [-0.35, 0.18] | 0.12 |
| **-F-R** | *SP '20* | -0.07 [-0.25, 0.12] | - | - | - | 0.46 |
| | *SU '20* | - | 0.02 [-0.30, 0.35] | -0.13 [-0.46, 0.16] | -0.18 [-0.49, 0.09] | 0.11 |
| | *SP '21* | - | 0.10 [-0.24, 0.44] | 0.07 [-0.35, 0.30] | 0.06 [-0.34, 0.26] | 0.74 |
| | *SU '21* | - | 0.07 [-0.23, 0.38] | 0.07 [-0.29, 0.29] | -0.06 [-0.40, 0.14] | 0.12 |

## 3.2 Direct and indirect effects of land-use and environmental factors on plant species richness and Shannon diversity in different seasons

We tested the direct and indirect effects (via light availability and alternatively via soil moisture) of fertilization and biomass removal on species richness by using a piecewise SEM for both spring and summer separately (Table 2). In spring, the SEM fitted species richness (Fisher's C = 15.135, df = 14, p = 0.369, n = 105, S2 Table) and Shannon diversity (Fisher's C = 17.263, df = 14, p = 0.242, n = 105, S3 Table) well (Fig 3A and 3B). For the SEM model on

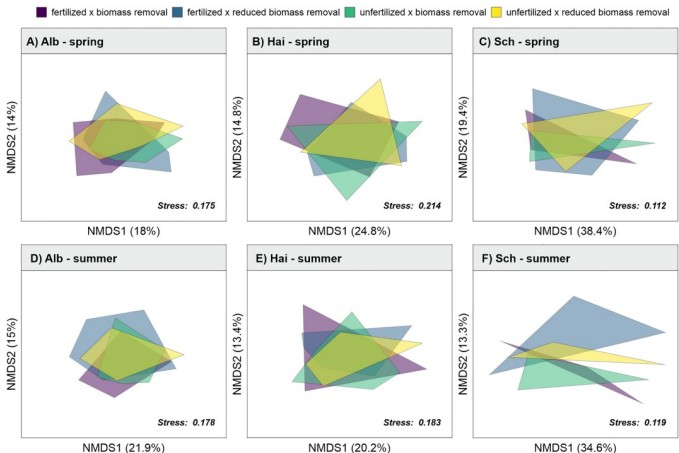

**Fig 2. Changes in community composition.** Non-metric multidimensional scaling (NMDS) based on Bray–Curtis similarity index for the plant communities for each treatment combination in spring (A-C) and summer (D-F) for 2021 (A,D) Schwäbische Alb, (B,E) Hainich-Dün, (C,F) Schorfheide Chorin. Hull volumes represent clusters of plant communities within a given treatment. Treatments are colour coded as fertilized & biomass removal (+F+R): purple; fertilized & reduced biomass removal (+F-R): bright blue, unfertilized & biomass removal (-F+R): green; unfertilized & reduced biomass removal (-F-R): yellow. Detailed model summaries are shown in the Supporting Information (S7 Table).

**Table 2. Selected standardized partial effect sizes of direct and indirect effects via light availability and soil moisture of fertilization or biomass removal on species richness and Shannon diversity with proposed interpretations.** Indirect effects were calculated by multiplying respective direct path coefficients. Direction of predicted (Pred) and observed (Obs) relationships are indicated as "pos." if positive, "neu." if neutral (if effect is smaller than +/- 0.005), and "neg." if negative. Bold text indicates whether effect was found to be significant. For further information see S3 and S6 Tables.

| | Effect | Mediator | Season | Magnitude | | Pred | Obs | Proposed interpretation |
|---|---|---|---|---|---|---|---|---|
| | | | | Rich | Div | | | |
| **Fertilization** | Direct | - | Spring | -0.061 | -0.075 | *neg.* | *neg.* | Contrary to our hypothesis (1), fertilization did not significantly affect plant species richness or Shannon diversity, neither directly nor indirectly, potentially because species gains as a consequence of a cessation in fertilization may need more time to emerge. |
| | | | Summer | -0.125 | -0.141 | | *neg.* | |
| | Indirect | Light availability | Spring | -0.006 | -0.008 | *neg.* | *neg.* | |
| | | | Summer | 0.001 | 0.002 | | *neu.* | |
| | | Soil moisture | Spring | -0.007 | -0.006 | | *neg.* | |
| | | | Summer | -0.001 | -0.001 | | *neu.* | |
| **Biomass removal** | Direct | - | Spring | -0.060 | -0.082 | *pos.* | *neg.* | In line with our hypothesis (2), biomass removal enhanced plant species richness/ Shannon diversity directly, supposedly via reducing soil nitrogen pools and preventing litter accumulation. Effects of biomass removal are largest in summer, potentially due to the accumulation of litter in the course of the growing season. |
| | | | Summer | 0.205 | 0.276 | *pos.* | **pos.** | |
| | Indirect | Light availability | Spring | 0.022 | 0.029 | *pos.* | *pos.* | Contrary to our hypothesis (2), biomass removal decreases Shannon diversity indirectly, by increasing light availability especially in summer, potentially due to lower drought-induced plant survival. However, as alternatively hypothesized, due to increases in soil moisture, mediated through changes in standing biomass, biomass removal positively affected species richness and Shannon diversity, likely due to decreased water stress. |
| | | | Summer | -0.066 | -0.110 | | **neg.**[a] | |
| | | Soil moisture | Spring | 0.024 | 0.023 | | *pos.* | |
| | | | Summer | 0.029 | 0.028 | | **pos.**[a] | |

[a] mixed effects found for species richness and Shannon diversity. Here only significant effects are reported, but see S3 and S6 Tables.

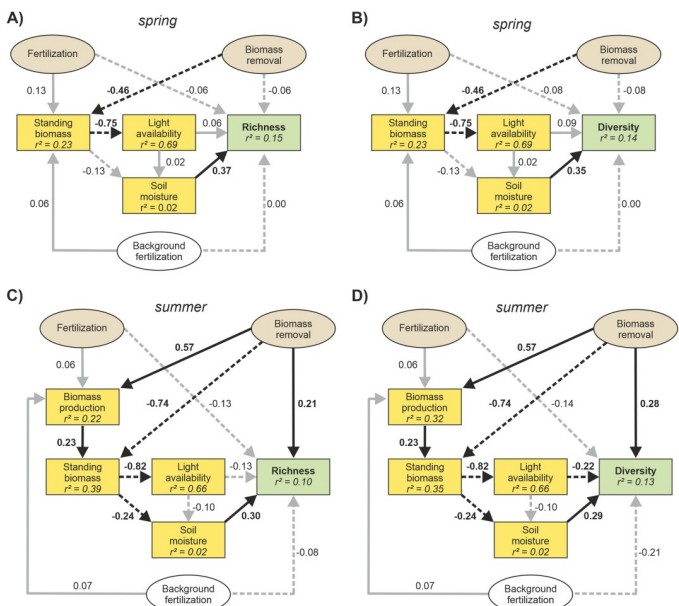

**Fig 3. Structural equation models with richness and diversity as main response.** Structural equation models for testing the direct and indirect effects of different land-use drivers (fertilization and biomass removal) on species richness in (A) spring (Fischer's C = 15.135, p = 0.369) and (C) summer (Fischer's C = 24.063, p = 0.344), and on Shannon diversity in (B) spring (Fischer's C = 17.263, p = 0.242), and (D) summer (Fischer's C = 26.006, p = 0.251) of both 2020 and 2021. Solid lines represent positive relationships and dotted lines negative relationships. Each arrow connection among variables indicates standardized path coefficients and significant levels (***p < 0.001, **p < 0.01, *p < 0.05). Detailed model summaries with unstandardized path coefficients are shown in the Supporting Information (S3 and S6 Tables).

the spring data, we found no evidence for direct or indirect (via light availability or soil moisture) causal pathways between fertilization or biomass removal and species richness/Shannon diversity. We found a strong partial effect between standing biomass (hereafter simply 'biomass') and light availability in the understory of the vegetation (direct path coeff. = −0.75). However, we did not find any significant relationship between light availability and species richness/Shannon diversity. In contrast, soil moisture was found to be directly positively related to species richness (direct path coeff. = 0.37) and Shannon diversity (direct path coeff. = 0.34). Additionally, we found that biomass removal strongly reduced the accumulation of biomass (direct path coeff. = −0.46), although no significant direct or indirect pathways between biomass removal and species richness or Shannon diversity were found. Overall, the SEM models in spring explained only 15% of the variance observed in species richness and 14% of the variance in species diversity. The most important predictor for species richness in spring was soil moisture (partial $r^2$ = 0.16), followed by fertilization (partial $r^2$ = 0.005), light availability (partial $r^2$ = 0.005) and biomass removal (partial $r^2$ = 0.004). For Shannon diversity, the most important predictor was soil moisture (partial $r^2$ = 0.142), followed by light availability (partial $r^2$ = 0.008), biomass removal (partial $r^2$ = 0.008) and fertilization (partial $r^2$ = 0.008). An NMDS analysis of the spring data showed that the effects of soil moisture on species richness and Shannon diversity cannot be explained by changes in species composition (Fig 4, S9 Table).

In summer, the SEM fitted species richness (Fisher's C = 24.063, df = 22, p = 0.344, n = 117, S5 Table) and Shannon diversity (Fisher's C = 26.006, df = 22, p = 0.251, n = 117, S6 Table) well (Fig 3C and 3D), including an additional error correlation between biomass production and soil moisture (species richness model: AIC without correlated error = 93.3, with correlated error = 92.1; Shannon diversity model: AIC without correlated error = 95.1, with correlated error = 94.0). For the SEM considering species richness and Shannon diversity, we found clear evidence for a positive, direct causal pathway of biomass removal on species richness/Shannon diversity (direct path coeff. = 0.21, direct path coeff. = 0.28, respectively, Table 1). We found only evidence for an indirect causal pathway between biomass removal and Shannon diversity,

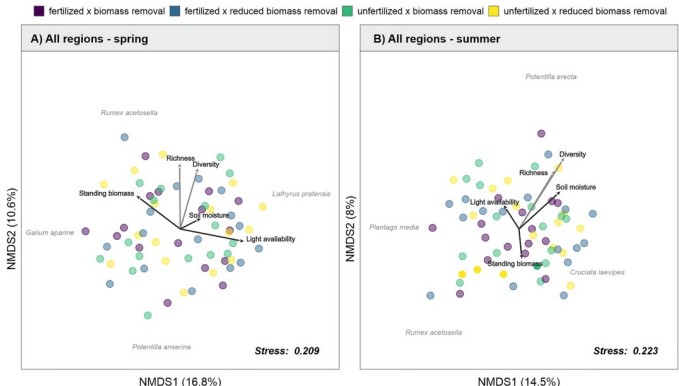

**Fig 4. Community composition and environmental factors.** Non-metric multidimensional scaling (NMDS) based on Bray–Curtis similarity index for the plant communities for each treatment combination and all regions combined in spring (A) and summer (B) for 2021. $R^2$-values are shown for each axis. Arrows printed in grey represent environmental factors: standing biomass, light availability and soil moisture, while arrows printed in red represent species richness and Shannon diversity. Species names represent the most extreme species according to the NMDS axes. Treatments are colour coded as fertilized & biomass removal (+F+R): purple; fertilized & reduced biomass removal (+F-R): bright blue, unfertilized & biomass removal (-F+R): green; unfertilized & reduced biomass removal (-F-R): yellow. Detailed model summaries are shown in the Supporting Information (S9 Table).

not species richness, mediated via changes in light availability (Fig 3, S5 Table). However, we did find evidence for an indirect causal pathway between biomass removal and species richness as well as Shannon diversity via soil moisture (Fig 3, S5 Table). For both models considering species richness or Shannon diversity, biomass removal led to an increase in biomass production (direct path coeff. = 0.57), while at the same time reducing standing biomass (direct path coeff. = -0.74). Furthermore, the accumulation of standing biomass led to a strong decrease in light availability (direct path coeff. = −0.82) and soil moisture (direct path coeff. = −0.24). We also found strong partial effects of biomass production on standing biomass (direct path coeff. = 0.23). However, we did not detect any strong direct or indirect effects of fertilization or background fertilization intensity on neither species richness, Shannon diversity (Table 1) nor biomass production. Overall, the SEM models in summer explained 11% of the variance observed in species richness and 13% of the variance in Shannon diversity. The most important predictor for species richness in summer was soil moisture (partial $r^2$ = 0.104), followed by biomass removal (partial $r^2$ = 0.043), fertilization (partial $r^2$ = 0.025), light availability (partial $r^2$ = 0.017) and background fertilization intensity (partial $r^2$ = 0.008). For Shannon diversity, the most important predictor was soil moisture (partial $r^2$ = 0.075), followed by biomass removal (partial $r^2$ = 0.057), background fertilization intensity (partial $r^2$ = 0.042), light availability (partial $r^2$ = 0.035) and fertilization (partial $r^2$ = 0.024). The NMDS analysis of the summer data showed that the positive effect of soil moisture on species richness in the summer season can be partly explained by changes in species composition (Fig 4, S9 Table), with moist plots being positively associated with both species richness, diversity and the first two NMDS axes, which represented a shift from communities showing higher abundances of *Rumex acetosella* in dry plots towards a higher abundance of *Potentilla erecta* in plots with a relatively high soil moisture.

## 4 Discussion

The present study aimed to advance our understanding of the complex relationship between land-use and plant biodiversity, by statistically and experimentally separating the effect of land-use drivers on plant biodiversity. Across three regions in Germany we found varying direct and indirect effects of fertilization and biomass removal on plant biodiversity depending on seasons and biodiversity facet (species richness and Shannon diversity). Although effects were generally weak in spring, in summer, biomass removal enhanced species richness and Shannon diversity, due to direct and partly indirect positive effects which compensated for negative effects (direct and indirect) of fertilization. Thus, our results support the hypothesis that biomass removal, such as mowing and grazing, can have compensatory effects on the generally negative effects of fertilization on plant biodiversity in managed grasslands.

To better understand the multivariate links between land-use drivers, observed plant species, as well as plant species richness and Shannon diversity in managed grasslands, we used structural equation modelling to test an integrated causal hypothesis (Fig 1). Although we did not observe strong changes in plant community composition in the different land-use treatments, our structural equation model revealed both direct and indirect effects of land-use on both species richness and Shannon diversity. We hypothesized that fertilization controls species richness and Shannon diversity both directly (e.g. by eutrophication, see [66]) and indirectly, via increased light competition resulting from increased standing biomass, [8, 12] or alternatively, through changes in soil moisture [14]. However, we did not detect strong direct or indirect effects of fertilization on species richness or Shannon diversity in spring and summer, while (non-significant) effects were mainly negative (albeit weakly), as expected. One possible explanation for the lack of strong direct or indirect effects of fertilization on species

richness or Shannon diversity might be that species gains in response to cessation of fertilization need more time to emerge, so that only small differences between fertilized and unfertilized treatments are visible in the short term [67, 68]. In general, light limitation as a consequence of fertilization-induced increases in standing biomass is suggested to negatively affect plant species richness and Shannon diversity, by changing competitive abilities of plants, consequentially favouring fast-growing and tall species [12, 19, 20]. However, in the present study we did not observe strong changes in the plant community composition. Only the plant community composition in Schorfheide-Chorin showed a tendency to differ between the land-use treatments. A possible explanation for the weak changes in plant community composition in response to the different land-use treatments is that even when reducing the intensity of certain land-use drivers, residual land-use effects (e.g. from former fertilization) might still be present in the system and disappear only in the long term [69].

Contrary to the weak negative effects of fertilization that we observed, biomass removal (by mowing or grazing) can promote species richness and Shannon diversity under certain conditions [25, 26, 30, 31]. We hypothesized that positive effects of biomass removal on species richness or Shannon diversity potentially compensate for the negative direct and indirect effects of fertilization. In general, and consistent with findings from previous studies [13, 70], biomass removal strongly decreased the accumulation of standing biomass in both seasons, having its greatest effect in summer. Furthermore, we observed that biomass removal increased biomass production, which has also been shown in previous studies [71], likely due to compensatory regrowth. However, direct effects of biomass removal on species richness or diversity differed strongly between seasons. In spring, biomass removal did not significantly affect species richness and Shannon diversity (although non-significant, weak negative effects were observed), while direct effects in summer were found to be significantly positive. Previous studies have found that early-season biomass removal events reduce species richness [6], likely explaining the negative (although weak and non-significant) relationships between biomass removal and both richness and Shannon diversity observed in spring. A possible explanation for the strong direct effects of biomass removal in summer might be that the removal of biomass prevented the accumulation of litter over the growing season [42]. Specifically, previous studies found that litter accumulation strongly affects community composition. This happens not only via changes in light availability but likely also via changes in soil acidity, nutrient mineralization or increased activity of pathogens, consequently having negative effects on species richness and diversity [15, 16]. Another possible mechanism by which biomass removal may have increased plant diversity is through the removal of nutrients, although this should only play a large role in mown (as opposed to grazed) grasslands [42, 43].

Similar to direct effects of biomass removal, indirect effects differed between seasons, with larger effects being generally found in summer compared to spring, suggesting varying importance of environmental variables across the growing season. We hypothesized that increased light availability would promote plant species richness and Shannon diversity, for example due to increased germination and seedling establishment [33, 72, 73]. However, in spring, contrary to our expectations, we found that such indirect effects were absent. Nevertheless, we did find that species richness and Shannon diversity were positively affected by soil moisture in spring, although soil moisture was not driven by biomass removal. A possible explanation for the lack of light mediated effects of land-use on both plant biodiversity in spring, might be that during the start of the growing season, the vegetation canopy is still relatively open and thus light is no limited resource. Another explanation for the general weak relationships between altered light availability (due to experimental changes in land-use drivers) and plant biodiversity observed in spring might be associated with the delayed impact of competitive release on

germination, which likely requires longer term observations to be detected [73]. In general, biodiversity change following a forcing event (such as land-use change), may only emerge in the long term, because of delayed immigration events of new species (immigration credit) or delayed extinction events (extinction dept) [74]. Hence, over time, we could expect an increasing importance of indirect effects of land-use on plant biodiversity in plots with reduced biomass removal due to an extinction debt (due to delayed extinction). Similarly, we could expect increased plant biodiversity only to be visible at the longer term, in plots with reduced fertilization intensity, due to an immigration credit.

In contrast to the observed relationships in spring, we found significant indirect effects of biomass removal on Shannon diversity in summer mediated through changes in light availability (negatively affecting diversity) as well as soil moisture (positively affecting diversity). Species richness though, was only significantly positively affected through biomass removal-induced changes in soil moisture. Contrary to our results, previous studies observed a positive relationship between light availability and species richness [8, 13]. A possible explanation for our contrasting results might be that contrary to our original hypothesis, in summer, soil moisture is more limiting for plant growth than light availability. Furthermore, the unexpected negative relationships between light availability and Shannon diversity could be explained by an increase in micro-temperatures, that might lead to increasing levels of transpiration especially in the summer season, potentially decreasing plant diversity [36, 37]. The positive indirect effect of biomass removal on species richness and Shannon diversity in summer mediated through changes in soil moisture, might be explained by short term effects of biomass removal. Thus, biomass removal may promote soil moisture by increasing rainfall recharge in upper soil layers due to reduced vegetation canopy coverage and root water consumption [75], although increased evaporation due to soil exposure to air can also have negative effects on soil moisture under certain conditions [76]. In general, disturbance is suggested to release resources, not previously available in the system [77]. Thus, in line with previous findings [78], our results suggest that biomass removal may indirectly promote plant species richness and Shannon diversity due to increased soil moisture in managed grasslands. Further, we found that the positive effect of biomass removal mediated through soil moisture on species richness and Shannon diversity in summer, could also be partly explained by shifts in the community composition. For example *Potentilla erecta (L.) Raeusch.*, was slightly associated with moist plots, while dryer plots were slightly associated with *Rumex acetosella L.* However, as our experiment only aimed to manipulate the intensity of land-use drivers, we cannot fully unravel the relative importance of land-use mediated changes in light availability in comparison to soil moisture and how that affects plant biodiversity. In the present study, the effects of both mowing and grazing were summarized mainly as effects of biomass removal. However, although both mowing and grazing have been found to promote plant biodiversity in grasslands [25, 26, 30, 31], the underlying mechanisms may differ, with grazing causing more patchy biomass removal, while mowing having more uniform effects [79]. As most of the grassland sites studied here were managed as meadows, it is possible that direct and indirect effects of biomass removal on plant biodiversity observed in this study were mostly driven by effects of mowing. Thus, to truly understand the complex interaction between different land-use drivers on plant biodiversity in managed grasslands it is necessary to not only disentangle the effects of fertilization and biomass removal, but also account for the type of biomass removal.

This study builds on previous studies, aiming to statistically and experimentally separate the effects of different drivers of land-use on plant biodiversity in managed grasslands [6, 12, 80–83]. However, contrary to these previous studies, we used high land-use intensity as a

default in comparison to treatments in which the intensity of certain land-use drivers was reduced (i.e. cessation in fertilization, and a reduction in biomass removal). This enabled us to disentangle the short-term direct and indirect effects of reduced fertilization and biomass removal in managed grasslands. We found that indirect effects of land-use on plant biodiversity were mainly driven by biomass removal, in particular in summer, mediated through both light availability and soil moisture. Specifically, the importance of soil moisture for plant biodiversity found in this study supports findings from previous studies [14, 38], suggesting that indirect effects of land-use mediated through light availability may depend on other environmental factors (such as soil moisture) that limit plant growth. Importantly, as climate change scenarios predict increasing temperatures in the future [84], the relative importance of indirect pathways of land-use mediated through changes in soil moisture might even increase. Overall, we did not find that indirect effects through changes in standing biomass were of greater importance than direct effects of land-use, in particular biomass removal. While in the present short-term study, we could not prove that a cessation in fertilization improves plant biodiversity, we did show that in these studied managed grasslands with likely high soil nutrient loads, decreasing biomass removal through a reduction in grazing intensity or mowing frequency negatively affects plant diversity. However, the relatively small number of sites and years investigated in this study, still limits insights into potential lag effects of land-use drivers on plant biodiversity. Thus, further research is needed to disentangle the complex interaction of different land-use drivers in managed grasslands. Additionally, as our experiment covered a relatively small gradient of land-use intensity, future studies should investigate the interactive effects of different land-use drivers on plant biodiversity along a wider gradient of land-use, while further comparing direct and indirect short vs long term land-use mediated effects. Nevertheless, this study may help us to understand how different land-use drivers interactively affect and thereby control biodiversity, which is crucial for informing biodiversity conservation.

## Supporting information

**S1 Fig. Overview showing the full factorial design.** In all three regions, four grasslands in the Schorfheide-Chorin, six grasslands in the Schwäbische Alb, and 6 grasslands in the Hainich-Dün (16 in total) were selected, where background land-use varied in mowing frequency, grazing intensity and fertilizer input. Within those grasslands, we established four 7×7 m treatments as a full factorial design with: fertilized & biomass removal, fertilized & reduced biomass removal (grazing stopped, but one late cut), unfertilized & biomass removal, unfertilized & reduced biomass removal. In all plots a 1×1 m subplot was established, where we measured plant biomass and performed vegetation surveys. Plot and subplot location was partly randomized in the grassland.
(DOCX)

**S2 Fig. Species richness across treatments.** Response of species richness ($m^{-2}$) on the fertilized & biomass removal (+F+R), unfertilized & biomass removal (-F+R), unfertilized & reduced biomass removal (-F-R) and fertilized & reduced biomass removal (+F-R) treatment for all three regions (Alb: Schwäbische Alb; Sch: Schorfheide-Chorin; Hai: Hainich-Dün). Whiskers correspond to the first and third quartiles. Treatments are colour coded as fertilization & biomass removal: purple; fertilized & reduced biomass removal: bright blue, unfertilized & biomass removal: green; unfertilized & reduced biomass removal: yellow. We did not observe any clear significant differences between any of the treatments (S2 Table).
(DOCX)

**S3 Fig. Shannon diversity per treatment.** Response of Shannon diversity (m-2) on the fertilized & biomass removal (+F+R), unfertilized & biomass removal (-F+R), unfertilized & reduced biomass removal (-F-R), and fertilized & reduced biomass removal (+F-R) treatment for all three regions (Alb: Schwäbische Alb; Sch: Schorfheide-Chorin; Hai: Hainich-Dün). Whiskers correspond to the first and third quartiles. Treatments are colour coded as fertilization & biomass removal: purple; fertilized & reduced biomass removal: bright blue, unfertilized & biomass removal: green; unfertilized & reduced biomass removal: yellow. Bars sharing a letter (a,b) do not differ significantly ($p < 0.05$, S10 Table). We only detected significant differences between treatments in summer 2020 within the Schwäbische Alb, thus no letters are shown for other years, seasons or regions.
(DOCX)

**S4 Fig. Standing biomass per treatment.** Response of standing biomass (g m$^{-2}$) on the fertilized & biomass removal (+F+R), unfertilized & biomass removal (-F+R), unfertilized & reduced biomass removal (-F-R), and fertilized & reduced biomass removal (+F-R) treatment for all three regions (Alb: Schwäbische Alb; Sch: Schorfheide-Chorin; Hai: Hainich-Dün). Whiskers correspond to the first and third quartiles. Treatments are colour coded as fertilization & biomass removal: purple; fertilized & reduced biomass removal: bright blue, unfertilized & biomass removal: green; unfertilized & reduced biomass removal: yellow. Bars sharing a letter (a,b) do not differ significantly ($p < 0.05$, S12 Table).
(DOCX)

**S5 Fig. Biomass production per treatment.** Response of biomass production (g m$^{-2}$) on the fertilization & biomass removal (+F+R), unfertilized & biomass removal (-F+R), unfertilized & reduced biomass removal (-F-R), and fertilized & reduced biomass removal (+F-R) treatment for all three regions (Alb: Schwäbische Alb; Sch: Schorfheide-Chorin; Hai: Hainich-Dün). Treatments are colour coded as fertilization & biomass removal: purple; fertilized & reduced biomass removal: bright blue, unfertilized & biomass removal: green; unfertilized & reduced biomass removal: yellow. Bars sharing a letter (a,b) do not differ significantly ($p < 0.05$, S14 Table).
(DOCX)

**S6 Fig. Correlation between land-use components.** Correlation between background LUI Index components, background fertilization intensity (kg N m-3 year-1), background mowing intensity (cuts year-1) and background grazing intensity (Livestock units * d ha-1 year-1) averaged across the years 2017–2019 and combined for all regions.
(DOCX)

**S7 Fig. Correlation between variables of SEM in spring.** Correlation matrix plot for the relationships between richness and Shannon diversity as well as the response variables standing biomass, light availability, and background fertilization intensity for spring of both 2020 and 2021, and colour coded for each region separately (Alb: Schwäbische Alb (red); Sch: Schorfheide-Chorin (blue); Hai: Hainich-Dün (green)).
(DOCX)

**S8 Fig. Correlation between variables of SEM in summer.** Correlation matrix plot for the relationships between richness and Shannon diversity as well as the response variables standing biomass, biomass production, light availability, and background fertilization intensity for summer of both 2020 and 2021, and colour coded for each region separately (Alb: Schwäbische Alb (red); Sch: Schorfheide-Chorin (blue); Hai: Hainich-Dün (green)).
(DOCX)

**S9 Fig. Partial residual plots, showing the effect of light availability and soil moisture on richness in spring (A,B) and summer (C,D), as well as on Shannon diversity in spring (E,F) and summer (G,H).** Partial residuals were extracted from model predicting species richness or Shannon diversity of respective SEM model. For more details on model estimates and significance see S3–S6 Tables.
(DOCX)

**S10 Fig. Changes in community composition in 2020.** Non-metric multidimensional scaling (NMDS) based on Bray–Curtis similarity index for the plant communities for each treatment combination in spring (A-C) and summer (D-F) 2020 (A,D) Schwäbische Alb, (B,E) Hainich-Dün, (C,F) Schorfheide Chorin. Hull volumes represent clusters of plant communities within a given treatment. Treatments are colour coded as fertilized & biomass removal (+F+R): purple; fertilized & reduced biomass removal (+F-R): bright blue, unfertilized & biomass removal (-F+R): green; unfertilized & reduced biomass removal (-F-R): yellow. Detailed model summaries are shown in the Supporting Information (S15 Table).
(DOCX)

**S11 Fig. Community composition and environmental factors in 2020.** Non-metric multidimensional scaling (NMDS) based on Bray–Curtis similarity index for the plant communities for each treatment combination and all regions combined in spring (A) and summer (B) for 2020. $R^2$-values are shown for each axis. Arrows printed in grey represent environmental factors: standing biomass, light availability and soil moisture, while arrows printed in red represent species richness and Shannon diversity. Species names represent the most extreme species according to the NMDS axes. Treatments are colour coded as fertilized & biomass removal (+F+R): purple; fertilized & reduced biomass removal (+F-R): bright blue, unfertilized & biomass removal (-F+R): green; unfertilized & reduced biomass removal (-F-R): yellow. Detailed model summaries are shown in the Supporting Information (S16 Table).
(DOCX)

**S1 Table. Species richness in response to treatments.** Linear mixed effect model showing the effect of the unfertilized & reduced biomass removal (-F-R), fertilized & reduced biomass removal (+F-R), unfertilized & biomass removal (-F+R) on species richness in comparison with the fertilized & biomass removal treatment for each regions (Alb: Schwäbische Alb; Sch: Schorfheide-Chorin; Hai: Hainich-Dün), as well as for different years and seasons.
(DOCX)

**S2 Table. Pairwise comparison of species richness across treatments.** Pairwise comparisons of the species richness in the fertilization & biomass removal (+F+R), unfertilized & reduced biomass removal (-F-R), unfertilized & biomass removal (-F+R) and fertilized & reduced biomass removal (+F-R) treatment, for each region (Alb: Schwäbische Alb; Sch: Schorfheide-Chorin; Hai: Hainich-Dün), as well as for different years and seasons. Significant (< 0.05) contrasts are written in bold.
(DOCX)

**S3 Table. PiecewiseSEM model fit for model with main response species richness in spring of both 2020 and 2021 with the main responses biomass removal (unfertilized & biomass removal treatment), fertilization (fertilized & reduced biomass removal treatment), standing biomass, log(light availability), background fertilization, region, sampling date and year.** Explained variances for species richness: mar. $R^2 = 0.15$ (adj. $R^2 = 0.40$), standing biomass: mar. $R^2 = 0.23$ (adj. $R^2 = 0.31$); log(light availability): mar. $R^2 = 0.69$ (adj. $R^2 = 0.69$); soil

moisture: mar. $R^2$ = 0.02 (adj. $R^2$ = 0.61). Alb: Schwäbische Alb; Sch: Schorfheide-Chorin; Hai: Hainich-Dün.
(DOCX)

**S4 Table. PiecewiseSEM model fit for model with main response Shannon diversity in spring of both 2020 and 2021 with the main responses biomass removal (unfertilized & biomass removal treatment), fertilization (fertilized & reduced biomass removal treatment), standing biomass, log(light availability), background fertilization region, sampling date and year.** Explained variances for Shannon diversity: mar. $R^2$ = 0.14 (adj. $R^2$ = 0.35), standing biomass: mar. $R^2$ = 0.23 (adj. $R^2$ = 0.31); log(light availability): mar. $R^2$ = 0.69 (adj. $R^2$ = 0.69); soil moisture: mar. $R^2$ = 0.02 (adj. $R^2$ = 0.61). Alb: Schwäbische Alb; Sch: Schorfheide-Chorin; Hai: Hainich-Dün.
(DOCX)

**S5 Table. PiecewiseSEM model fit for model with main response species richness in summer of both 2020 and 2021 with the main responses biomass removal (unfertilized & biomass removal treatment), fertilization (fertilized & reduced biomass removal treatment), standing biomass, biomass production, log(light availability), background fertilization region, sampling date and year.** Explained variances for species richness: mar. $R^2$ = 0.11 (adj. $R^2$ = 0.56), standing biomass: mar. $R^2$ = 0.35 (adj. $R^2$ = 0.69); biomass production: mar. $R^2$ = 0.32 (adj. $R^2$ = 0.43); log(light availability): mar. $R^2$ = 0.66 (adj. $R^2$ = 0.67); soil moisture: mar. $R^2$ = 0.02 (adj. $R^2$ = 0.74). Alb: Schwäbische Alb; Sch: Schorfheide-Chorin; Hai: Hainich-Dün.
(DOCX)

**S6 Table. PiecewiseSEM model fit for model with main response Shannon diversity in summer of both 2020 and 2021 with the main responses biomass removal (unfertilized & biomass removal treatment), fertilization (fertilized & reduced biomass removal treatment), standing biomass, biomass production, log(light availability), background fertilization region, sampling date and year.** Explained variances for Shannon diversity: mar. $R^2$ = 0.10 (adj. $R^2$ = 0.36), standing biomass: mar. $R^2$ = 0.35 (adj. $R^2$ = 0.69); biomass production: mar. $R^2$ = 0.32 (adj. $R^2$ = 0.43); log(light availability): mar. $R^2$ = 0.66 (adj. $R^2$ = 0.67); soil moisture: mar. $R^2$ = 0.02 (adj. $R^2$ = 0.74). Alb: Schwäbische Alb; Sch: Schorfheide-Chorin; Hai: Hainich-Dün.
(DOCX)

**S7 Table. Pairwise comparison (PERMANOVA) of plant community composition per treatment in 2021 between the fertilized & biomass removal (+F+R), unfertilized & reduced biomass removal (-F-R), fertilized & reduced biomass removal (+F-R), unfertilized & biomass removal (-F+R) treatments for all three regions (Alb: Schwäbische Alb; Sch: Schorfheide-Chorin; Hai: Hainich-Dün) and for both spring and summer 2021 (see Fig 3).**
(DOCX)

**S8 Table. Permutation test of fitted vectors of the environmental variables (standing biomass, light availability and soil moisture), plant species richness and Shannon diversity in 2021 on the NMDS ordination (NMDS1 and NMDS2) for spring and summer of 2021 across all regions (Schwäbische Alb, Hainich-Dün, Schorfheide-Chorin) (S5 Fig).**
(DOCX)

**S9 Table. Shannon diversity in response to treatments.** Linear mixed effect model showing the effect of the unfertilized & reduced biomass removal (-F-R), fertilized & reduced biomass removal (+F-R), unfertilized & biomass removal (-F+R) on Shannon diversity in comparison

with the fertilized & biomass removal (+F+R) treatment for each regions (Alb: Schwäbische Alb; Sch: Schorfheide-Chorin; Hai: Hainich-Dün), as well as for different years and seasons. (DOCX)

**S10 Table. Pairwise comparison of Shannon diversity across treatments.** Pairwise comparisons of the Shannon diversity in the fertilization & biomass removal (+F+R), unfertilized & reduced biomass removal (-F-R), unfertilized & biomass removal (-F+R) and fertilized & reduced biomass removal (+F-R) treatment, for each region (Alb: Schwäbische Alb; Sch: Schorfheide-Chorin; Hai: Hainich-Dün), as well as for different years and seasons. No pairwise comparison shown for spring 2021, as the predictor 'treatment' was not part of the most parsimonious model. Significant (< 0.05) contrasts are written in bold. (DOCX)

**S11 Table. Standing biomass in response to treatments.** Linear mixed effect model showing the effect of the unfertilized & reduced biomass removal (-F-R), fertilized & reduced biomass removal (+F-R), unfertilized & biomass removal (-F+R) on standing biomass in comparison with the fertilized & biomass removal treatment for each regions (Alb: Schwäbische Alb; Sch: Schorfheide-Chorin; Hai: Hainich-Dün), as well as for different years and seasons. (DOCX)

**S12 Table. Pairwise comparison of standing biomass across treatments.** Pairwise comparisons of the standing biomass in the fertilization & biomass removal, unfertilized & reduced biomass removal, unfertilized & biomass removal and fertilized & reduced biomass removal treatment, for each region (Alb: Schwäbische Alb; Sch: Schorfheide-Chorin; Hai: Hainich-Dün), as well as for different years and seasons. Significant (< 0.05) contrasts are written in bold. Due to missing data on fertilized & biomass removal treatments in spring 2020 for the Schorfheide-Chorin no pairwise contrasts shown (*). (DOCX)

**S13 Table. Biomass production in response to treatments.** Linear mixed effect model showing the effect of the unfertilized & reduced biomass removal (-F-R), fertilized & reduced biomass removal (+F-R), unfertilized & biomass removal (-F+R) on biomass production in comparison with the fertilized & biomass removal treatment for each regions (Alb: Schwäbische Alb; Sch: Schorfheide-Chorin; Hai: Hainich-Dün), in summer for all different years. (DOCX)

**S14 Table. Pairwise comparison of biomass production across treatments.** Pairwise comparisons of biomass production in the fertilization & biomass removal (+F+R), unfertilized & reduced biomass removal (-F-R), unfertilized & biomass removal (-F+R) and fertilized & reduced biomass removal (+F-R) treatment, for each region (Alb: Schwäbische Alb; Sch: Schorfheide-Chorin; Hai: Hainich-Dün), as well as for different years and seasons. Significant (< 0.05) contrasts are written in bold. Due to missing data on fertilized & biomass removal treatments in spring 2020 for the Schorfheide-Chorin no pairwise contrasts shown (*). (DOCX)

**S15 Table. Pairwise comparison (PERMANOVA) of plant community composition per treatment in 2020 between the fertilized & biomass removal (+F+R), unfertilized & reduced biomass removal (-F-R), fertilized & reduced biomass removal (+F-R), unfertilized & biomass removal (-F+R) treatments for all three regions (Alb: Schwäbische Alb; Sch: Schorfheide-Chorin; Hai: Hainich-Dün) and for both spring and summer 202O (see**

**S9 Fig).**
(DOCX)

**S16 Table. Permutation test of fitted vectors of the environmental variables (standing biomass, light availability and soil moisture), plant species richness and Shannon diversity in 2020 on the NMDS ordination (NMDS1 and NMDS2) for spring and summer of 2020 across all regions (Schwäbische Alb, Hainich-Dün, Schorfheide-Chorin) (S10 Fig).**
(DOCX)

**S17 Table. Mean LUI drivers (grazing, mowing and fertilization) averaged across 2017 to 2019 for each region separately (Alb: Schwäbische Alb, Hai: Hainich-Dün, Sch: Schorfheide-Chorin).**
(DOCX)

**S18 Table. Calibration model of biomass estimates based on a linear regression between actual standing biomass measurements (dry matter in g m$^{-2}$) and rising plate meter measurement (1/2 cm increments), combined for all regions.** Calibration was performed for data measured in spring 2021.
(DOCX)

**S1 File.**
(TXT)

**S2 File.**
(RMD)

# Acknowledgments

We thank Svenja Kunze, Ralph Bolliger, Uta Schumacher, Jörg Hailer, Christin Schreiber, Victoria Henning and many students for helping us to collect data in the field. We also thank Christian Wirth and Christiane Roscher for their input during the conceptual phase of this paper. Also many thanks to the managers of the three Exploratories, Max Müller, Robert Künast, Franca Marian, and all former managers for their work in maintaining the plot and project infrastructure; Victoria Grießmeier for giving support through the central office, Andreas Ostrowski for managing the central database, and Markus Fischer, Eduard Linsenmair, Dominik Hessenmöller, Ingo Schöning, François Buscot, Ernst-Detlef Schulze, Wolfgang W. Weisser and the late Elisabeth Kalko for their role in setting up the Biodiversity Exploratories project. We thank the administration of the Hainich national park, the UNESCO Biosphere Reserve Swabian Alb and the UNESCO Biosphere Reserve Schorfheide-Chorin as well as all land owners for the excellent collaboration. Field work permits were issued by the responsible state environmental offices of Baden-Württemberg, Thüringen, and Brandenburg (according to § 72 BbgNatSchG).

# Author Contributions

**Conceptualization:** Alexandra Weigelt, Daniel Prati, Fons van der Plas.

**Data curation:** Karl Andraczek, Judith Hinderling, Fons van der Plas.

**Formal analysis:** Karl Andraczek, Fons van der Plas.

**Funding acquisition:** Fons van der Plas.

**Investigation:** Karl Andraczek, Lena Kretz, Daniel Prati, Fons van der Plas.

**Methodology:** Karl Andraczek, Alexandra Weigelt, Judith Hinderling, Lena Kretz, Daniel Prati, Fons van der Plas.

**Project administration:** Alexandra Weigelt, Fons van der Plas.

**Resources:** Alexandra Weigelt, Judith Hinderling, Fons van der Plas.

**Supervision:** Alexandra Weigelt, Judith Hinderling, Daniel Prati, Fons van der Plas.

**Validation:** Alexandra Weigelt, Fons van der Plas.

**Visualization:** Karl Andraczek.

**Writing – original draft:** Karl Andraczek, Fons van der Plas.

**Writing – review & editing:** Karl Andraczek, Alexandra Weigelt, Judith Hinderling, Lena Kretz, Daniel Prati, Fons van der Plas.

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
