## [Editor Report · Decision Letter 0]

23 Jan 2023

PONE-D-22-34453Biomass removal promotes plant diversity after short-term extensification of managed grasslandsPLOS ONE

Dear Dr. Andraczek, Thank you for submitting your manuscript to PLOS ONE. After careful consideration, we feel that it has merit but does not fully meet PLOS ONE’s publication criteria as it currently stands. Therefore, we invite you to submit a revised version of the manuscript that addresses the points raised during the review process.

And thanks for you patience with the review process - I got sick with covid (still positive - yay) - which explains some of the delay. But more importantly, multiple reviewers declined to review the paper and rather than continuing to chase down more at this time, I suggest that figures be processed to make them more legible. If I received this as a reviewer, I would say "nope - can't read figures." I also suggest some minor edits that may help. So "Major Revision" here is intended to flag that it will go to reviewers (vs reject [a go away] or minor revision [editor could handle it]).

Please see below for details, and then resubmit with me suggested as Academic Editor again, including a quick summary in your cover letter of how you approached the list below. I will then try again on the reviewer-hunting process, with the hope that edits will make it more amenable to reviewers.

My comments:

1. What is the difference between intensification and extensification? If different, please explain. If actually the same, then I suggest using “intensification” consistently.

2. Must the experiment be justified in a BEF context? Why not simply a matter of understanding land use intensification effects on biodiversity? In other words, the BEF context may not be needed and only adds an extra layer of inference here, whereas biodiversity seems worthy of preserving regardless. Think about it.

3. Shannon diversity: please define with its equation. It could be the original H’, but as Jost (2006) nicely demonstrated, that is an entropy that requires exp(H’) to obtain effective diversity. Thus “Shannon diversity” has taken on several meanings of late. I suggest calculating and using Jost’s effective diversity, which corresponds to Hill number 1 (1D).

4. Fig. 1 Are the numbers related to a sequence (implied by numbering)? If not, perhaps they can be removed. Also, I suggest more obviously dashed lines for negative effects: on the pdf the resolution is too low, meaning that dashes are not clearly different from the solid lines – especially when zoomed out to be comparable to the size of a figure in a printed journal page (imagine figures squeezed into a column or ½ width of paper).

5. MOST IMPORTANTLY: Resolution of all Figures in the received pdf (and what reviewers see) is too low, so that text in figures is illegible and shapes are very indistinct. I expect negative reviews based only on this technical problems rather than merits of the work. According to the PLOS One office, "It appears the reason this error is occurring is because the authors provided figure files that are incompatible with Editorial Manager's PDF compilation. We encourage authors to use the the PACE tool (https://pacev2.apexcovantage.com/) on all of their Fig files to ensure the images will be of the highest quality in PDF form." Thus I strongly suggest following those PLOS figure guidelines for a revised submission.

A letter that responds to each point raised above. You should upload this letter as a separate file labeled 'Response to Reviewers'.A new copy of your manuscript with improved figures and edits for my comments (as you see fit). You should upload this as a separate file labeled 'Revised Manuscript with Track Changes'.An unmarked version of your revised paper without tracked changes. You should upload this as a separate file labeled 'Manuscript'. This will go to reviewers.

Guidelines for resubmitting your figure files are available below, at the end of this letter.

We look forward to receiving your revised manuscript.

Kind regards,

David G. Jenkins, PhD

Academic Editor

PLOS ONE

Journal Requirements:

"The work has been funded by the German Research Foundation (DFG) Priority Program 1374 ‘Biodiversity-Exploratories’ (BEF-Loops Nr.433266560, Exploratories project phase Nr. PL 891/3-1)."

5. Please include a caption for figure 2. 

[NOTE: The submitted manuscript is attached to this email and accessible via the submission site. Please log into your account, locate the manuscript record, and check for the action link "View Attachments". If this link does not appear, there are no attachment files.]

---

## [Author Response · Author response to Decision Letter 0]

10 Feb 2023

Editor’s comments

We very much appreciate that the editor finds our study interesting and for giving us a chance to resubmit the manuscript. Regarding the specific comments by the editor, we refer to our responses on the more detailed comments below.

#1. What is the difference between intensification and extensification? If different, please explain. If actually the same, then I suggest using “intensification” consistently.

We thank the editor for this useful suggestion on clarifying the difference between ‘intensification’ and ‘extensification’. As a short answer here: with land-use extensification we are referring to the opposite of land-use intensification, hence a reduction in land-use intensity (i.e. reduction in mowing frequency, grazing or fertilization intensity). We now provided a definition in the introduction to improve clarity about the terminologies used (see changes in lines 56-57). 

#2. Must the experiment be justified in a BEF context? Why not simply a matter of understanding land use intensification effects on biodiversity? In other words, the BEF context may not be needed and only adds an extra layer of inference here, whereas biodiversity seems worthy of preserving regardless. Think about it. 

We originally included the BEF context to stress the importance of why we need to advance our understanding on how land-use drives biodiversity loss, as this loss may have detrimental consequences for ecosystem functioning. However, as the editor correctly points out, that rather adds an additional layer of complexity to our study, while biodiversity in itself is worthy of preserving regardless. Thus, we agree to the editor’s suggestion that our introduction would benefit from refining our focus and now excluded the link to the BEF context (the sentences on lines 51-53 of our original manuscript has been removed). 

#3. Shannon diversity: please define with its equation. It could be the original H’, but as Jost (2006) nicely demonstrated, that is an entropy that requires exp(H’) to obtain effective diversity. Thus “Shannon diversity” has taken on several meanings of late. I suggest calculating and using Jost’s effective diversity, which corresponds to Hill number 1 (1D). 

Following this suggestion, we now adjusted all our analyses by using exp(H’) instead of just H’. We now also provided the specific formula used to calculate effective diversity in the methods section (see changes in lines: 210-212). While most of the results did not change after using the exp(H’), as the only qualitative difference we now find that indirect effects of biomass removal on diversity were not only mediated via changes in light availability, but also via soil moisture (Fig. 4, Table 1; lines 369-371, 489-493). Hence, using effective diversity revealed the relative importance of soil moisture in summer as an indirect pathway mediated by biomass removal, similar to what we observed for species richness, making our findings more consistent.

#4. Fig. 1 Are the numbers related to a sequence (implied by numbering)? If not, perhaps they can be removed. Also, I suggest more obviously dashed lines for negative effects: on the pdf the resolution is too low, meaning that dashes are not clearly different from the solid lines – especially when zoomed out to be comparable to the size of a figure in a printed journal page (imagine figures squeezed into a column or ½ width of paper). 

Yes, the numbers within the figure are related to a sequence of supporting references for each path within the causal framework and hence, we would prefer to keep the numbers within the figure. We now clarified this issue in line 108. The specific references are included in the figure caption. 

#5. Also, I suggest more obviously dashed lines for negative effects: on the pdf the resolution is too low, meaning that dashes are not clearly different from the solid lines – especially when zoomed out to be comparable to the size of a figure in a printed journal page (imagine figures squeezed into a column or ½ width of paper). MOST IMPORTANTLY: Resolution of all Figures in the received pdf (and what reviewers see) is too low, so that text in figures is illegible and shapes are very indistinct. I expect negative reviews based only on this technical problems rather than merits of the work. According to the PLOS One office, "It appears the reason this error is occurring is because the authors provided figure files that are incompatible with Editorial Manager's PDF compilation. We encourage authors to use the the PACE tool (https://pacev2.apexcovantage.com/) on all of their Fig files to ensure the images will be of the highest quality in PDF form." Thus I strongly suggest following those PLOS figure guidelines for a revised submission.

We agree, and apologize for the low figure quality of the figures we provided. We now increased the quality of all figures (and ensured journal requirements for all figures using the PACE tool) within the document and, to better visualize negative of positive hypothesised relationships, also selected a different line-type for the dashed lines.

---

## [Decision Letter · Decision Letter 1]

29 Mar 2023

PONE-D-22-34453R1Biomass removal promotes plant diversity after short-term extensification of managed grasslandsPLOS ONE

Dear Dr. Andraczek,

Thank you for submitting your manuscript to PLOS ONE. After careful consideration, we feel that it has merit but does not fully meet PLOS ONE’s publication criteria as it currently stands. Therefore, we invite you to submit a revised version of the manuscript that addresses the points raised during the review process. You will see that we received commens from one outside expert reviewer, which I supplemented with my own detailed review, Both are written in the spirit of improving the manuscript toward publication, and I encourage you to read and respond to each comment with your revisions so that a decision can be made promptly on your revised manuscript. I anticipate handling that phase without sending it out for external review again, given that my comments may lead to more revisions than the reviewer's, and that the next decision should not be too difficult.

We look forward to receiving your revised manuscript.

Kind regards,

David G. Jenkins, PhD

Academic Editor

PLOS ONE

Additional Editor Comments:

Dear Dr. Andraczek and co-authiors,

Below you will find I sought at least two reviewers but in the end obtained only one. Thus I also wrote comments aimed to improve the manuscript – in the way a reviewer should. Please do not assign any more weight to my comments below in your responses and revisions than you would to an anonymous reviewer (i.e., feel free to rebut my thoughts with good arguments).

I find Reviewer 1 is generally complementary about the manuscript and provides good suggestions. As usual, the reviewer and I see different features to emphasize – a good thing! I especially agree with the reviewer in the suggestion that the short-term effects should be emphasized, which would be consistent with my comments below about potential, and acknowledged, lag effects. Reviewer 1 suggested Minor Revisions, but I think the aggregate of their comments and mine add up to Major Revision (for what that’s worth). I also think that having resolved or dispelled comments here, the manuscript should be ready to accept.

Please see the details on Reviewer 1’s comments via the information provided by PLOS One. Below are my comments on the manuscript. Finally, thanks for your patience in this review process.

Sincerely,

David G. Jenkins

Title and lines 56-57 and elsewhere. The word “extensification” is not the opposite of “intensification” and is not used by the cited papers [9, 10]. If I look it up in online dictionaries I see definitions like this:

1. the process of making something (more) extensive

2. the geographic spread and distribution of any technology, especially agriculture.

Thus I suggest it be replaced with an antonym, such as one of those listed at https://www.thesaurus.com/browse/intensification? Or perhaps "mitigation"? Or more clumsy: "de-intensification"?

Line 247. I expected SEM software etc. to be listed in this section. It may help to re-organize subheadings?

Line 255. I strongly encourage authors to move beyond repeated pairwise Tukey tests and rely more on GLMs and SEMs for interpreting statistical trends.

Paragraph starting at line 292. I strongly recommend the authors use AIC.psem in the piecewise SEM package to compare models for “efficiency” or “plausibility.” This will provide a more robust test of which model to spend more time with than other tests, and then leaves goodness of fit measures as a way to criticize the most efficient model selected by AIC. See extensive literature by Burnham and Anderson for more details on AICs. The AIC results should then be a table in the manuscript, followed by results for the "winning" model.

Fig. 2 revealed little to me. I recommend the authors instead show and discuss a table of GLM output from lme4: are coefficients for biomass removal and fertilizer addition clearly non-zero? What are the signs? What is the overall fit? Speaking of regression output, S1 Table is close to what I sought, but I think it reveals errors that matter:

• Why is is Spring 2020 different from other time steps in its model structure?

• The intercept (default by R) is Alb but also +F + R. It may make more sense to choose (using relevel in R) to use Alb but -F -R for a control condition as the intercept. Also I suggest converting SE to 95% confidence intervals (1.96 X SE) and omitting the DF and t value columns. And note that rows are relative to the intercept.

Finally, the study was designed as if site and time could be random effects, though with too few levels of each to really help as random effects. That makes all the sites and times show as separate estimates. The minimum number of levels needed for random effects in a mixed effect model may not be met here (that threshold is not clear), but I suggest trying a GLMM instead of the GLM. That would potentially move sites and time into random effects, leaving main effects of interest as fixed effects and making interpretation more straight-forward. I may be wrong - it may not work because sample size is limiting. If so, please just say so in your response to review comments. But - if it works, it may be more efficient and effective as a test of treatment effects.

Line 313. This sentence could be edited to be more direct. Beyond that, this section 4.1 based on regression could be argued (above?) as seeking overall (i.e., direct and indirect) effects of treatments on diversity, whereas SEM seeks to tease apart the direct from the indirect. Thus regression evaluates overall / net effects while SEM evaluates details. Perhaps that can help reconcile the greater information in Fig. 4 than in Fig. 2. And by the way, the use of SEM is an advance to make clear for this research subject – much research has not used it but work here shows it helps bring clarity.

Line 417. I don’t know what is meant by “land use reductions.”

Paragraph starting at Line 425 and subsequent paragraphs. It is usually expected that Figures are not cited in Discussion, to ensure that Results are clearly separate from Discussion. If it feels like Figures need to be used, then the sentence may belong better in Results.

Line 438. Yes – I agree; time lags are often observed, and some citations to support this would be useful here.

Paragraph starting at Line 448. This paragraph covers almost 3 pages! Organize and subdivide.

Line 522. Careful! Not only is “extensification”used here again, but I think the authors need to be careful to avoid claiming a first. Consult the rich literature by Tilman and a small army of others at Cedar Creek. In addition, other studies that used a broader range of fertilizer and biomass removal have explored hump-shaped patterns related to the Intermediate Disturbance Hypothesis, Intermediate Productivity Hypothesis, and Huston’s combination of those two ideas (his Dynamic Equilibrium Model). There is a long history of inquiry here that includes factors here as part of the story, as the co-authors well know (they have been involved in some). Among others, a few that may be relevant include:

• Feng-Wei Xu, Jian-Jun Li, Li-Ji Wu, Xiao-Ming Lu, Wen Xing, Di-Ma Chen, Biao Zhu, Shao-Peng Wang, Lin Jiang, Yong-Fei Bai, Resource enrichment combined with biomass removal maintains plant diversity and community stability in a long-term grazed grassland, Journal of Plant Ecology, 113(5), October 2020, Pages 611–620, https://doi.org/10.1093/jpe/rtaa046

• Band, N., Kadmon, R., Mandel, M. and DeMalach, N., 2022. Assessing the roles of nitrogen, biomass, and niche dimensionality as drivers of species loss in grassland communities. Proceedings of the National Academy of Sciences, 119(10), p.e2112010119.

• Carnus, T., Connolly, J., Kirwan, L. and Finn, J.A., 2013. Plant diversity effects are robust to cutting severity and nitrogen application in productive grasslands. In The role of grasslands in a green future: threats and perspectives in less favoured areas. Proceedings of the 17th Symposium of the European Grassland Federation, Akureyri, Iceland, 23-26 June 2013 (pp. 195-197). Agricultural University of Iceland.

I think the concluding paragraph could benefit from a sentence or two on limits of the study – as difficult as it is to get this much data, it still has limits in the number places and times, for strong analyses and to capture acknowledged potential lag effects.

Reviewers' comments:

Reviewer's Responses to Questions

**Comments to the Author**

1. If the authors have adequately addressed your comments raised in a previous round of review and you feel that this manuscript is now acceptable for publication, you may indicate that here to bypass the “Comments to the Author” section, enter your conflict of interest statement in the “Confidential to Editor” section, and submit your "Accept" recommendation.

Reviewer #1: (No Response)

2. Is the manuscript technically sound, and do the data support the conclusions?

Reviewer #1: Yes

3. Has the statistical analysis been performed appropriately and rigorously? 

Reviewer #1: Yes

4. Have the authors made all data underlying the findings in their manuscript fully available?

Reviewer #1: Yes

5. Is the manuscript presented in an intelligible fashion and written in standard English?

Reviewer #1: Yes

6. Review Comments to the Author

Reviewer #1: The authors present results on plant biodiversity, community composition and plant biomass from an in-field experiment in which the land-use components fertilization and biomass removal have been reduced in a full-factorial design. The results represent the short-term effects one and two years after the experiments have been implemented. Although effects on the plant community are very small overall, the study does provide interesting insights into possible mechanistic pathways as the authors also included light availability and soil moisture in their models. Both the experimental design and the statistical analyses are done rigorously and correctly, hence the conclusions are supported by the data.

* The main strength of the experiment compared to previous studies is the implementation of all treatments within the same grassland sites. This could be emphasised more in the introduction.

* The introduction lacks an explanation of the expected "seasonal variability of relationships" mentioned in the legend of Figure 1. As spring and summer data are analysed and discussed separately, a section on seasonal differences should be added in the introduction.

* The (indirect) effect of biomass production on diversity is mentioned in the hypotheses, but not explained and motivated in the Introduction. As biomass production is an important component in the model structure, the expected effects and mechanisms should be added to the introduction.

* The experiments were set up in grasslands of different land-use intensity. While the average intensity values are given in the Supplement, the description in the Methods section is inconsistent: In line 142f "a gradient in land-use intensity" is mentioned, while in line 163 management is described as "relatively high" and in line 163 as "medium to high". In addition to making the statements consistent, I recommend describing the range of land-use components on the grasslands by the actual number and time of cuts, amount of fertilizer and/or number of grazing animals (and not just in terms of the standardized land-use intensity).

* The description of the treatments is in some parts difficult to read, especially if several treatments are mentioned in one sentence. I would prefer the use of abbreviations as done for the supplementary tables. The authors could also consider calling the treatment with fertilizer and biomass removal the control.

* The terms subplot, plot and field are used inconsistently in the first part of the manuscript. See for instance line 176f. Please use consistent terminology and I would recommend not using "field" but rather "grassland".

Minor comments:

Line 146: "the 10 most abundant"

Line 171: Why was the fertilizer manually applied in this treatment? Why was is not fertilized together with the "control" treatment and the remaining grassland?

Line 173f: This information can be moved to the next section on field data collection

Line 158: What does "setting-up" mean exactly? Does this mean that the plots were marked or was e.g. fertilization in autumn 2019 already reduced? Please clarify in which year/season the treatments were applied for the first time.

Line 243 ff: This sentence contains the same information twice.

Line 254: remove "with"

Line 257: which additional predictors are those?

Line 313 ff: there is no need to mention the type of treatment again in the parentheses

Line 350f (and later): it is not quite clear what the "respectively" refers to. The path coefficients should be given directly after richness and diversity.

Figure 2: the y-axis label seems incorrect (remove the minus)

7. PLOS authors have the option to publish the peer review history of their article (what does this mean?). If published, this will include your full peer review and any attached files.

Reviewer #1: No

---

## [Author Response · Author response to Decision Letter 1]

8 May 2023

Editor’s comments

I find Reviewer 1 is generally complementary about the manuscript and provides good suggestions. As usual, the reviewer and I see different features to emphasize – a good thing! I especially agree with the reviewer in the suggestion that the short-term effects should be emphasized, which would be consistent with my comments below about potential, and acknowledged, lag effects. Reviewer 1 suggested Minor Revisions, but I think the aggregate of their comments and mine add up to Major Revision (for what that’s worth). I also think that having resolved or dispelled comments here, the manuscript should be ready to accept. Please see the details on Reviewer 1’s comments via the information provided by PLOS One. Below are my comments on the manuscript. Finally, thanks for your patience in this review process.

We very much appreciate the complementary comments provided by both the editor and the reviewer. We fully agree with the general recommendation to emphasize short-term and lag effects and adjusted the respective sections in our discussion (see changes in lines: 39, 126-128, 451-454, 541, 557-558). Regarding the specific comments by the editor and the reviewers, we refer to our responses to the more detailed comments below.

Title and lines 56-57 and elsewhere. The word “extensification” is not the opposite of “intensification” and is not used by the cited papers [9, 10]. If I look it up in online dictionaries I see definitions like this: 1. the process of making something (more) extensive 2. the geographic spread and distribution of any technology, especially agriculture. Thus I suggest it be replaced with an antonym, such as one of those listed at https://www.thesaurus.com/browse/intensification? Or perhaps "mitigation"? Or more clumsy: "de-intensification"? 

We thank the editor for this comment and now used the term “de-intensification” when referring to reduction of land-use intensity throughout the manuscript and within our title (“Biomass removal promotes plant diversity after short-term de-intensification of managed grasslands”). See also lines 1-2, 56, 110, 127 and 154.

Line 247. I expected SEM software etc. to be listed in this section. 

We agree and now added the SEM software in line 257.

It may help to re-organize subheadings? 

We agree and re-organized and re-named the subheadings in the methods section to allow a better distinction between “conventional” statistical methods (linear mixed effect models and pairwise contrasts) and more in-depth analyses using multivariate analyses (NMDS) as well as piecewiseSEM. Specifically, we refined subheadings in the methods (see lines: 258, 272, 284) and result section (see lines: 324).

Line 255. I strongly encourage authors to move beyond repeated pairwise Tukey tests and rely more on GLMs and SEMs for interpreting statistical trends.

We fully agree with the editor that GLMs and SEMs provide more in-depth statistical results compared to analyses based on pairwise Tukey tests. We noticed that we communicated this wrongly in our original manuscript and apologize for the confusion: 

In our original manuscript we did not use the Tukey HSD test (as incorrectly mentioned in our methods section) to compute pairwise differences. Rather, after testing for overall effects of treatments within our Linear Mixed Model, we applied the more advanced emmeans function from the package “emmeans”, which computes contrasts between treatments based on estimated least-square means. To clarify this information we corrected all respective description about pairwise tests throughout our manuscript (see lines: 261-263, 327-328). 

In our manuscript we deliberately analysed differences between treatments using “conventional” Linear Mixed Models as a first step, to test for treatment effects. However, we also agree with the editor that more advanced statistical methods are needed to provide more insights into the mechanisms (e.g. through changes in vegetation biomass [and hence light competition], or changes in soil moisture) by which these treatments influence plant biodiversity indicators. Hence, in our manuscript we used piecewiseSEM for this purpose (methods: section 3.5.3, results: section 4.2).

Paragraph starting at line 292. I strongly recommend the authors use AIC.psem in the piecewise SEM package to compare models for “efficiency” or “plausibility.” This will provide a more robust test of which model to spend more time with than other tests, and then leaves goodness of fit measures as a way to criticize the most efficient model selected by AIC. See extensive literature by Burnham and Anderson for more details on AICs. The AIC results should then be a table in the manuscript, followed by results for the "winning" model.

We thank the editor for this useful suggestion and now compared the respective models using AIC.psem from the piecewise SEM package. Accordingly, in the methods we now explained that we compared the AICs of each model when alternative model structures were tested (see lines: 304-311, 375-379). 

Specifically, we started with constructing a priori models (separately for richness and diversity in spring and summer) based on our hypothetical framework introduced in our paper. If alternative pathways improved model fit, we now additionally compared competing models based on the AIC. This was only done in summer, as there, an error correlation between biomass production and soil moisture improved overall model fit. Hence, we reported these AIC results in our manuscript. However, as model fit of our a priori developed spring models was already good, we did not compare any competing models with the spring models, in line with the philosophy that ideally, only a priori developed models in SEMs are shown, unless model fit suggests that initial models are not adequate yet (see lines: 306-311). 

Fig. 2 revealed little to me. I recommend the authors instead show and discuss a table of GLM output from lme4: are coefficients for biomass removal and fertilizer addition clearly non-zero? What are the signs? What is the overall fit? 

We agree, and now put Figure 2 of the original manuscript into our supplement, and additionally provided a table showing the standardized effect sizes of the treatment effects on species richness in the main manuscript (see Table 1, lines 335-341). In line with the suggestions from the editor, this table now more clearly shows the sign, confidence intervals and overall fit for each model. 

Speaking of regression output, S1 Table is close to what I sought, but I think it reveals errors that matter:

• Why is is Spring 2020 different from other time steps in its model structure?

The statistical output for the model testing the treatment effect on species richness in spring 2020 is different compared to the other seasons or years, as the most parsimonious model (based on AIC) did not contain an interaction between treatment and the respective regions. To emphasise this fact, we now provided some additional information in the table caption (see lines in supplement: 113-114). 

• The intercept (default by R) is Alb but also +F + R. It may make more sense to choose (using relevel in R) to use Alb but -F -R for a control condition as the intercept. 

While we agree that from a theoretical point of view, -F-R might be a more reasonable default, we deliberately choose +F+R as default, as in our experimental design the “unmanipulated” plots were +F+R (where land-use intensity reflected the local management by the respective farmer). Hence, we would prefer to keep +F+R as default in our models, unless the editor wishes otherwise. 

Also I suggest converting SE to 95% confidence intervals (1.96 X SE) and omitting the DF and t value columns. And note that rows are relative to the intercept.

We agree and now provided the 95% CI in all our tables, omitted the DF and t-values, and added the note that rows are relative to the intercept.

Finally, the study was designed as if site and time could be random effects, though with too few levels of each to really help as random effects. That makes all the sites and times show as separate estimates. The minimum number of levels needed for random effects in a mixed effect model may not be met here (that threshold is not clear), but I suggest trying a GLMM instead of the GLM. That would potentially move sites and time into random effects, leaving main effects of interest as fixed effects and making interpretation more straight-forward. I may be wrong - it may not work because sample size is limiting. If so, please just say so in your response to review comments. But - if it works, it may be more efficient and effective as a test of treatment effects.

We fully agree that a GLMM with regions and year as random effects would help to correct our estimates. However, unfortunately, our study design lacks the required number of levels (region only 3, year only 2) for these variables to be used as random effects (rough threshold of minimal 5 levels, see Chapter 11, Section 5 in “Data Analysis Using Regression and Mulitilevel/Hierarchical models” by Gelman and Hill, 2007). Hence, while appreciating the suggestion by the editor, a GLMM with random effects would unfortunately not improve our current modelling approach. 

Line 313. This sentence could be edited to be more direct. 

We agree, and reformulated and restructured this sentence to be more direct (see line: 326-329).

Beyond that, this section 4.1 based on regression could be argued (above?) as seeking overall (i.e., direct and indirect) effects of treatments on diversity, whereas SEM seeks to tease apart the direct from the indirect. Thus regression evaluates overall / net effects while SEM evaluates details. Perhaps that can help reconcile the greater information in Fig. 4 than in Fig. 2. And by the way, the use of SEM is an advance to make clear for this research subject – much research has not used it but work here shows it helps bring clarity.

We fully agree and appreciate the editors recognition of our use on more “advanced” statistical methods, such as SEM, in addition to “conventional” statistics, to distinguish between direct and indirect treatment effects. According to the editors suggestions, we now renamed subheadings in the methods section and in the results section, to clarify how the different statistical approaches used in our study answer different questions. We hope that this will also help the reader to reconcile the greater information in Fig. 2. 

Line 417. I don’t know what is meant by “land use reductions.”

We reformulated the sentence to increase clarity (see line: 431-433).

Paragraph starting at Line 425 and subsequent paragraphs. It is usually expected that Figures are not cited in Discussion, to ensure that Results are clearly separate from Discussion. If it feels like Figures need to be used, then the sentence may belong better in Results.

We agree with the suggestion of the editor and removed all references to Figures or Tables from the discussion. 

Line 438. Yes – I agree; time lags are often observed, and some citations to support this would be useful here.

We agree and now provided additional references on lag effects found in previous studies (see line: 454). 

Paragraph starting at Line 448. This paragraph covers almost 3 pages! Organize and subdivide.

We apologize for the inconvenient format of this paragraph and re-organized the respective section into three separate paragraphs (see re-organized paragraphs in lines: 464-538).

Line 522. Careful! Not only is “extensification” used here again, but I think the authors need to be careful to avoid claiming a first. Consult the rich literature by Tilman and a small army of others at Cedar Creek. In addition, other studies that used a broader range of fertilizer and biomass removal have explored hump-shaped patterns related to the Intermediate Disturbance Hypothesis, Intermediate Productivity Hypothesis, and Huston’s combination of those two ideas (his Dynamic Equilibrium Model). There is a long history of inquiry here that includes factors here as part of the story, as the co-authors well know (they have been involved in some). Among others, a few that may be relevant include:

• Feng-Wei Xu, Jian-Jun Li, Li-Ji Wu, Xiao-Ming Lu, Wen Xing, Di-Ma Chen, Biao Zhu, Shao-Peng Wang, Lin Jiang, Yong-Fei Bai, Resource enrichment combined with biomass removal maintains plant diversity and community stability in a long-term grazed grassland, Journal of Plant Ecology, 113(5), October 2020, Pages 611–620, https://doi.org/10.1093/jpe/rtaa046

• Band, N., Kadmon, R., Mandel, M. and DeMalach, N., 2022. Assessing the roles of nitrogen, biomass, and niche dimensionality as drivers of species loss in grassland communities. Proceedings of the National Academy of Sciences, 119(10), p.e2112010119.

• Carnus, T., Connolly, J., Kirwan, L. and Finn, J.A., 2013. Plant diversity effects are robust to cutting severity and nitrogen application in productive grasslands. In The role of grasslands in a green future: threats and perspectives in less favoured areas. Proceedings of the 17th Symposium of the European Grassland Federation, Akureyri, Iceland, 23-26 June 2013 (pp. 195-197). Agricultural University of Iceland.

We thank the editor for this important comment and apologize for the confusion. We fully agree, that there are many previous studies, investigating the effect of different land-use drivers on plant biodiversity. To clarify this, we now rephrased the respective sections in our discussion (see lines: 539-543), emphasizing that we build on knowledge of previous studies. Additionally, we now specifically refer to our experimental design (high land-use intensity as a default in comparison to treatments in which the intensity of certain land-use drivers was reduced) to emphasize the novelty of our study compared to previous research (see lines: 541-543). Finally, we very much appreciated the useful references provided and referred to them in our manuscript.

I think the concluding paragraph could benefit from a sentence or two on limits of the study – as difficult as it is to get this much data, it still has limits in the number places and times, for strong analyses and to capture acknowledged potential lag effects.

We fully agree, and now provided some additional information on the limitations of our study in the discussion section and highlighted avenues for future research (see lines: 557-558, 560-563).

Reviewer #1:

Reviewer #1: The authors present results on plant biodiversity, community composition and plant biomass from an in-field experiment in which the land-use components fertilization and biomass removal have been reduced in a full-factorial design. The results represent the short-term effects one and two years after the experiments have been implemented. Although effects on the plant community are very small overall, the study does provide interesting insights into possible mechanistic pathways as the authors also included light availability and soil moisture in their models. Both the experimental design and the statistical analyses are done rigorously and correctly, hence the conclusions are supported by the data.

We thank the reviewer for the positive feedback on our study. Regarding the specific comments by the reviewer, we refer to our responses to the more detailed comments below.

* The main strength of the experiment compared to previous studies is the implementation of all treatments within the same grassland sites. This could be emphasised more in the introduction.

We thank the reviewer for this suggestion and now emphasized our experimental treatment within the introduction (see lines: 102-105).

* The introduction lacks an explanation of the expected "seasonal variability of relationships" mentioned in the legend of Figure 1. As spring and summer data are analysed and discussed separately, a section on seasonal differences should be added in the introduction.

We agree and now added two sections providing further rational on why we expect “seasonal variability of relationships” in our introduction (see lines: 66-69, 88-90).

* The (indirect) effect of biomass production on diversity is mentioned in the hypotheses, but not explained and motivated in the Introduction. As biomass production is an important component in the model structure, the expected effects and mechanisms should be added to the introduction.

We now clarified our motivation why indirect effects of fertilization are mediated by changes in biomass production in our introduction (see lines: 63-66).

* The experiments were set up in grasslands of different land-use intensity. While the average intensity values are given in the Supplement, the description in the Methods section is inconsistent: In line 142f "a gradient in land-use intensity" is mentioned, while in line 163 management is described as "relatively high" and in line 163 as "medium to high". In addition to making the statements consistent, I recommend describing the range of land-use components on the grasslands by the actual number and time of cuts, amount of fertilizer and/or number of grazing animals (and not just in terms of the standardized land-use intensity).

We apologize for the inconsistency when describing the land-use intensity gradient at the given grasslands and now provided information on the range of the intensity of each land-use driver in our methods section (see lines: 171-173).

* The description of the treatments is in some parts difficult to read, especially if several treatments are mentioned in one sentence. I would prefer the use of abbreviations as done for the supplementary tables. The authors could also consider calling the treatment with fertilizer and biomass removal the control.

We appreciate the useful suggestion to use the treatment abbreviations and adjusted the treatment names throughout the manuscript. Within the figures, we only provided the full treatment names, while in the caption, we referenced both the full and the abbreviated treatment name. This was done to guarantee the connection between abbreviated and full treatment names, to make sure readers understand our figures even if they did not read the full manuscript. 

* The terms subplot, plot and field are used inconsistently in the first part of the manuscript. See for instance line 176f. Please use consistent terminology and I would recommend not using "field" but rather "grassland".

We fully agree and now clarified terminologies throughout the manuscript when referring to “plots” or “subplots”. Additionally, we now use “grassland” instead of “field” throughout the manuscript. 

Minor comments:

Line 146: "the 10 most abundant"

We agree and added quotation marks in line 156. 

Line 171: Why was the fertilizer manually applied in this treatment? Why was is not fertilized together with the "control" treatment and the remaining grassland?

The +F-R and the -F-R subplots were both located in a plot where land-use was reduced. While in the +F+R, the farmer applied fertilizer using heavy machinery (fertilizer is sprayed over relatively large area), fertilizer was manually applied in the +F-R subplots, to avoid cross-contamination between treatments. We clarified this information in the methods section, see lines: 179-182.

Line 173f: This information can be moved to the next section on field data collection

We agree and deleted the respective text from the “Experimental design” section.

Line 158: What does "setting-up" mean exactly? Does this mean that the plots were marked or was e.g. fertilization in autumn 2019 already reduced? Please clarify in which year/season the treatments were applied for the first time.

With “setting-up” we refer to plots, that were marked and, in some cases (+F-R and -F-R) fenced in autumn 2019. Reduction in both biomass removal or fertilization was started from autumn 2019 onwards. We specified this information and provided further clarification in our method section (see lines: 167-168).

Line 243 ff: This sentence contains the same information twice.

We now rephrased the respective sentence (see lines: 248-251).

Line 254: remove "with"

We adjusted respective sentence. 

Line 257: which additional predictors are those?

To test how species richness differed between treatments, we stepwise added “treatment” and “region”. The predictor “treatment” was always kept in the final model, irrespective of whether it was part of the most parsimonious model or not. We clarified this information in the methods section in line: 264-267, 269-271.

Line 313 ff: there is no need to mention the type of treatment again in the parentheses#

We agree and adjusted the respective sentences. 

Line 350f (and later): it is not quite clear what the "respectively" refers to. The path coefficients should be given directly after richness and diversity.

We agree and adjusted the respective sentences (see line: 362-364). 

Figure 2: the y-axis label seems incorrect (remove the minus)

Species richness was quantified as the number of vascular plant species within a 1x1 m subplot. Hence, we used the label “species richness (m-2)” to indicated that species richness is given per square meter.

---

## [Editor Report · Decision Letter 2]

29 May 2023

Biomass removal promotes plant diversity after short-term de-intensification of managed grasslands

PONE-D-22-34453R2

Dear Dr. Andraczek,

We’re pleased to inform you that your manuscript has been judged scientifically suitable for publication and will be formally accepted for publication once it meets all outstanding technical requirements. Thanks again for your patience in the review process, and thank you for handling review comments so well.

Kind regards,

David G. Jenkins, PhD

Academic Editor

PLOS ONE

---

## [Editor Report · Acceptance letter]

8 Jun 2023

PONE-D-22-34453R2 

Biomass removal promotes plant diversity after short-term de-intensification of managed grasslands 

Dear Dr. Andraczek:

I'm pleased to inform you that your manuscript has been deemed suitable for publication in PLOS ONE. Congratulations! Your manuscript is now with our production department. 

Kind regards, 

on behalf of

Dr. David G. Jenkins 

Academic Editor

PLOS ONE